# Clustering units in neural networks: upstream vs downstream information

**Richard D. Lange**                                                    *rdlange@seas.upenn.edu*
*Department of Neurobiology*
*University of Pennsylvania*
*Philadelphia, PA 19104*

**David S. Rolnick**
*Mila Québec AI Institute*
*McGill University*
*Montréal, Canada H3A 0G4*

**Konrad P. Kording**
*Department of Neurobiology*
*University of Pennsylvania*
*Philadelphia, PA 19104*

**Reviewed on OpenReview:** *https://openreview.net/forum?id=Euf7KofunK*

## Abstract

It has been hypothesized that some form of "modular" structure in artificial neural networks should be useful for learning, compositionality, and generalization. However, defining and quantifying modularity remains an open problem. We cast the problem of detecting functional modules into the problem of detecting *clusters* of similar-functioning units. This begs the question of what makes two units functionally similar. For this, we consider two broad families of methods: those that define similarity based on how units respond to structured variations in inputs ("upstream"), and those based on how variations in hidden unit activations affect outputs ("downstream"). We conduct an empirical study quantifying modularity of hidden layer representations of a collection of feedforward networks trained on classification tasks, across a range of hyperparameters. For each model, we quantify pairwise associations between hidden units in each layer using a variety of both upstream and downstream measures, then cluster them by maximizing their "modularity score" using established tools from network science. We find two surprising results: first, dropout dramatically increased modularity, while other forms of weight regularization had more modest effects. Second, although we observe that there is usually good agreement about clusters within both upstream methods and downstream methods, there is little agreement about the cluster assignments across these two families of methods. This has important implications for representation-learning, as it suggests that finding modular representations that reflect structure in inputs (e.g. disentanglement) may be a distinct goal from learning modular representations that reflect structure in outputs (e.g. compositionality).

## 1 Introduction

*Modularity* is a design principle in which systems that solve complex problems are decomposed into sub-systems that solve simpler problems and that can be independently analyzed, debugged, and recombined for new tasks. From an engineering perspective, modular design has many benefits, such as increased robustness and fast adaptation to new problems (Simon, 1962). It is well-known that learning systems benefit from

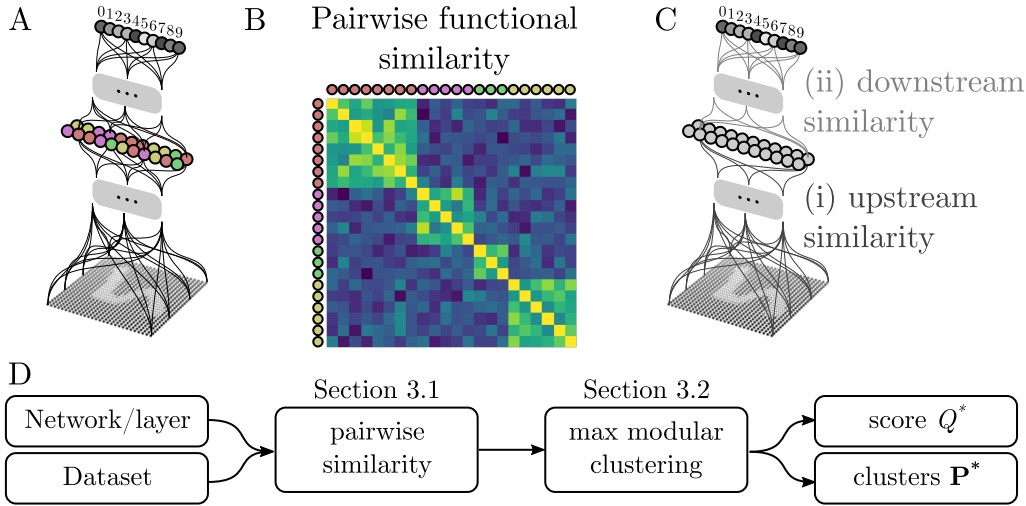

Figure 1: **A)** Given a neural network, such as this schematic of a fully-connected network for classifying MNIST digits, we frame the problem of detecting modules that perform distinct functions inside the network as a problem of identifying clusters of units in a hidden layer (colors). **B)** Given some method for quantifying the *pairwise* functional similarity between any two units, we operationalize a "module" as a subset of units with high intra-module functional similarity and low similarity to units in other modules, which appears as high block-diagonal values in the pairwise similarity matrix after sorting by cluster. **C)** In this setup, the main challenge is to define what makes any pair of two units "functionally similar" such that two units are "similar" if they are part of the same functional module. We distinguish between two broad classes of methods: (i) those that score similarity based on how "upstream" factors influence the pair, and (ii) those based on how the pair influences "downstream" behavior of the network. **D)** Schematic of our pipeline to quantify modularity: given a layer of a network, we compute the $n \times n$ matrix of "similarity" of all pairs of units in that layer. We then search for the maximally modular clustering such that the similarity matrix becomes heavily block-diagonal as in **B**. The output is a modularity score ($Q$) and cluster-assignments ($P$), for each layer and for each method of quantifying pairwise similarity.

structure adapted to the problem at hand (Denker et al., 1987), and many real-world problems indeed can be decomposed into sub-problems. It is no surprise, then, that modularity is considered a normative design principle of evolved biological systems (Wagner et al., 2007; Lipson, 2007), including in biological neural networks (Azam, 2000; Kashtan & Alon, 2005; Kashtan et al., 2007; Clune et al., 2013), and one from which artificial neural networks (ANNs) can benefit.

Despite these strong intuitions, formally defining and quantifying the modularity of a given system remains an open problem (cf. Simon (1962); Lipson (2007)). It is generally accepted that, by definition, modular systems decompose into sub-systems that implement functions that solve sub-problems. Characterizing modules in ANNs therefore begs the question: when are two parts of a network involved in the same "function" (Csordás et al., 2021)? In this paper, we ask this question at the level of pairs of hidden units. That is, we consider a variety of methods for measuring the "functional similarity" of any given pair of hidden units, and we operationalize a "module" as a cluster of similarly-functioning units (Figure 1A-B). This definition is not intended to solve the question of what a module is, but to provide a tractable basis for experimenting with various ideas related to modularity, such as how it is affected by regularization.

One goal of this paper is simply to raise awareness of qualitative differences between "upstream" and "downstream" ways of thinking about neural representations and the functions they are involved in (Figure 1C). In section 3, we make these definitions quantitative and give details for our method for detecting and quantifying functional modules in the hidden layers of trained neural networks by clustering units into similarly-functioning groups. This framework allows us to directly compare a variety of proxies for the amount of modularity a network exhibits. Experimental results are described in section 4. In addition to quantitatively scoring modularity, we further investigate whether different similarity measures agree on which units belong to which module. Surprisingly, we find that clusters discovered according to "upstream" measures of functional similarity are consistently distinct from those discovered using "downstream" measures. Although we do not consider regularization methods explicitly designed to produce modular designs, these initial empirical results nonetheless call for a deeper look at how the "function" of a representation is defined, as well as why and in which situations modules may be useful.

## 2 Related Work

The study of modularity in neural networks has a long history (Clune et al., 2013; Amer & Maul, 2019). One common source of inspiration from biology is the functional dissociation of "what" and "where" pathways in the ventral and dorsal streams of the brain, respectively (Goodale & Milner, 1992). Each of these pathways can be seen as a specialized module (and may be further decomposed into submodules), and many previous experiments on modularity in artificial neural networks have explored principles that would result in similarly distinct what/where information processing in ANNs (Rueckl et al., 1989; Jacobs et al., 1991; Di Ferdinando et al., 2001; Bakhtiari et al., 2021). A key difference to this line of work is that, rather than specifying the functional role of modules ahead of time, such as one module being "what" and another being "where," our work seeks to identify distinct functional clusters in trained networks.

Broadly speaking, there are two families of approaches to modularity in neural networks, corresponding to different ways of thinking about the function of parts of a network. The *structural modularity* family of approaches defines function in terms of network weights and how sub-networks connect to other sub-networks, and so modules are defined as densely intra- and sparsely inter-connected sub-networks (Watanabe et al., 2018; Amer & Maul, 2019; Csordás et al., 2021; Filan et al., 2021; Hod et al., 2021). The *functional modularity* family of approaches emphasizes network activations rather than weights, or the information represented by those activations. These include disentanglement, compositionality, invariance, and others(Bengio et al., 2013; Higgins et al., 2018; Andreas, 2019; Scholkopf et al., 2021; Watanabe, 2019; Watanabe et al., 2019; 2020). The relationship between structural and functional modules is not clear – while it certainly seems that they are (or ought to be) correlated (Lipson, 2007), it has been observed empirically that even extremely sparse inter-module-connectivity does not always guarantee functional separation of information-processing (Béna & Goodman, 2021). In this work, we take the functional approach, based on the assumption that structural modularity is itself only useful insofar as it supports distinct functions of the units, and that often distinct functions must share information, making strict structural delineations potentially counterproduc-

tive. For example, knowing "what" an object is in a challenging visual scene helps localize "where" it is, and vice versa.

Our work is perhaps most closely related to a series of papers by Watanabe and colleagues in which trained networks are decomposed into clusters of "similar" units with the aim of understanding and simplifying those networks. In some cases, they quantify the similarity of units structurally using a combination of both incoming and outgoing weights (Watanabe et al., 2018), and in other cases functionally using correlation between hidden unit activations (Watanabe, 2019; Watanabe et al., 2019; 2020). These are similar in spirit to our goal of identifying modules by clustering units, but an interesting contrast to our approach; while they combine "upstream" and "downstream" information for each pair of units, we find stark differences in results between the two.

## 3 Quantifying modularity by clustering similarity

We split the problem of identifying functional modules into two stages (Figure 1D): scoring the pairwise similarity of units, then clustering based on similarity. To simplify, we only apply these steps separately to each hidden layer, but in principle modules could be scored in the same way after concatenating layers together. Section 3.1 below defines the set of pairwise functional similarity methods we use, and Section 3.2 describes the clustering stage.

While we focus on similarity between pairs of individual units, our approach is related to, and inspired by, the question of what makes neural representations "similar" when comparing entire populations of neurons to each other (Kornblith et al., 2019). Rather than finding clusters-of-similar-neurons as we do here, one could instead define modules in terms of dissimilarity-between-clusters-of-neurons. In preliminary work, we explored such a definition of functional modules, using representational (dis)similarity between sub-populations of neurons. The main challenge of this approach is that existing representational-similarity methods are highly sensitive to dimensionality (the number of neurons in each cluster), and it is not obvious how best to control for this when computing dissimilarity between clusters such that the method is not biased to prefer large or small cluster sizes.[1] To further motivate our method, note that representational similarity analysis is closely related to tests for statistical (in)dependence between populations of neurons (Kornblith et al., 2019; Gretton et al., 2005; Cortes et al., 2012), and so the problem of finding mutually "dissimilar" modules is analogous to the problem of finding *independent subspaces* (Hyvärinen et al., 2001; Gutch & Theis, 2007). In Independent Subspace Analysis (ISA), there is an analogous problem of determining what constitutes a surprising amount of dependence between subspaces of different dimensions, and various methods have been proposed with different inductive biases (Wu et al., 2021; Bach & Jordan, 2004; 2003; Póczos & Lőrincz, 2005). However, Palmer & Makeig showed that a solution to the problem of detecting independent *subspaces* is to simply *cluster* the individual dimensions of the space. This provides some justification for the methods we use here: some technicalities[2] notwithstanding, the problem of finding subspaces of neural activity with "dissimilar" representations is, in many cases, reducible to the problem of *clustering* individual units based on pairwise similarity, as we do here.

### 3.1 Quantifying pairwise similarity of hidden units

What makes two hidden units "functionally similar"? In other words, we seek some similarity function $S : \mathbb{N}, \mathbb{D}, \mathbb{T} \rightarrow \mathbb{R}_+^{n \times n}$ that takes as input a neural network $\mathbb{N}$ evaluated on a dataset $\mathbb{D}$ to solve task $\mathbb{T}$ and produces an $n \times n$ matrix of non-negative similarity scores for all pairs among the $n$ hidden units. We further require that the resulting matrix is symmetric, or $S_{ij} = S_{ji}$. Importantly, allowing $S$ to depend on the task $\mathbb{T}$ opens the door to similarity measures where units are considered similar based on their downstream contribution to a particular loss function.

---

[1] Note that dimensionality is not a problem for other applications of representational (dis)similarity methods, since the population size is traditionally fixed in any single analysis.

[2] First, one can construct pathological examples of *dependent* subspaces but where all individual variables are *independent*, so pairwise independence between all elements in a cluster cannot guarantee independent clusters. Second, summing or averaging pairwise dissimilarity across clusters will give quantitatively different value than measuring dissimilarity between the populations, but this brings us back to the dimensionality problem already discussed.

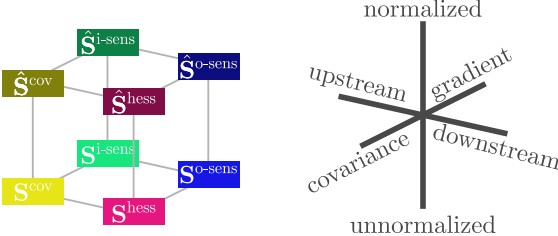

Figure 2: Color scheme and 2x2x2 visualization of the 8 proposed similarity methods.

**Similarity by covariance.** The first similarity measure we consider is the absolute value covariance of hidden unit activities across inputs. Let $\boldsymbol{x}_k \in \mathbb{R}^d$ be the $k$th input in the dataset, and $\boldsymbol{h}_i(\boldsymbol{x})$ be the response of the $i$th hidden unit to input $\boldsymbol{x}$, with $i \in \{1, 2, \ldots, n\}$. Then, we define similarity as

$$\boldsymbol{S}_{ij}^{\mathrm{cov}} = \frac{1}{K-1} \left| \sum_{k=1}^{K} \left( \boldsymbol{h}_i(\boldsymbol{x}_k) - \bar{\boldsymbol{h}}_i \right) \left( \boldsymbol{h}_j(\boldsymbol{x}_k) - \bar{\boldsymbol{h}}_j \right) \right| \tag{1}$$

where $K$ is the number of items in $\mathbb{D}$ and $\bar{\boldsymbol{h}}_i \equiv \frac{1}{K} \sum_{k=1}^{K} \boldsymbol{h}_i(\boldsymbol{x}_k)$ is the mean response of unit $i$ on the given dataset. Intuitively, the absolute value covariance quantifies the statistical dependence of two units across inputs, making it an **upstream** measure of similarity. $\boldsymbol{S}^{\mathrm{cov}}$ corresponds to some classic notions of "disentanglement": to the extent that the rows and columns of $\boldsymbol{S}^{\mathrm{cov}}$ can be sorted to have highly block-diagonal structure, this means that hidden units cluster into statistically independent – or at least uncorrelated – groups. This is also similar in spirit to Representational Similarity Analysis, where the similarity of *sets* of units is quantified using some (often kernel-based) measure of their independence on a given dataset (Kornblith et al., 2019).

**Similarity by input sensitivity.** While $\boldsymbol{S}^{\mathrm{cov}}$ quantifies similarity of responses *across* inputs, we next consider a measure of similar sensitivity to *single* inputs, which is then averaged over $\mathbb{D}$. Let $\boldsymbol{J}_{\boldsymbol{x}}^{\boldsymbol{h}}$ denote the $n \times d$ Jacobian matrix of partial derivatives of each hidden unit with respect to each of the $d$ input dimensions. Then, we say two units $i$ and $j$ are similarly *sensitive* to input changes on input $\boldsymbol{x}_k$ if the dot product between the $i$th and $j$th row of $\boldsymbol{J}_{\boldsymbol{x}_k}^{\boldsymbol{h}}$ has high absolute-value magnitude. In matrix notation over the entire dataset, we use

$$\boldsymbol{S}^{\mathrm{i\text{-}sens}} = \frac{1}{K} \left| \sum_{k=1}^{K} \boldsymbol{J}_{\boldsymbol{x}_k}^{\boldsymbol{h}} \boldsymbol{J}_{\boldsymbol{x}_k}^{\boldsymbol{h}\top} \right| \tag{2}$$

where the superscript "i-sens" should be read as the "input sensitivity."

**Similarity by last-layer sensitivity.** Let $\boldsymbol{y} \in \mathbb{R}^o$ denote the last-layer activity of the network. Using the same Jacobian notation as above, let $\boldsymbol{J}_{\boldsymbol{h}}^{\boldsymbol{y}}$ denote the $o \times n$ matrix of partial derivatives of the last layer with respect to changes in the hidden activities $\boldsymbol{h}$. Then, we define similarity by output sensitivity as

$$\boldsymbol{S}^{\mathrm{o\text{-}sens}} = \frac{1}{K} \left| \sum_{k=1}^{K} \boldsymbol{J}_{\boldsymbol{h}}^{\boldsymbol{y}\top} \boldsymbol{J}_{\boldsymbol{h}}^{\boldsymbol{y}} \right|, \tag{3}$$

likewise with "o-sens" to be read as "output-sensitivity." Note that both $\boldsymbol{h}$ and $\boldsymbol{y}$ depend on the particular input $\boldsymbol{x}_k$, but this has been left implicit in the notation to reduce clutter.

**Similarity by the loss Hessian.** The "function" of a hidden unit might usefully be thought of in terms of its contribution to the task or tasks it was trained on. To quote Lipson,

> "In order to measure modularity, one must have a quantitative definition of function... It is then possible to take an arbitrary chunk of a system and measure the dependency of

the system function on elements within that chunk. The more that the dependency *itself* depends on elements outside the chunk, the less the function of that chunk is localized, and hence the less modular it is." (Lipson, 2007).

Lipson then goes on to suggest that the "dependence of system function on elements" can be expressed as a derivative or gradient, and that the dependence *of that dependence* on other parts of the system can be expressed as the second derivative or Hessian. Towards this conception of modular functions on a particular task, we use the following definition of similarity:

$$S_{ij}^{\text{hess}} = \frac{1}{K} \left| \sum_{k=1}^{K} \frac{\partial^2 L}{\partial \boldsymbol{h}_i \partial \boldsymbol{h}_j} \right| , \tag{4}$$

where $L$ is the scalar loss function for the task, and should be understood to depend on the particular input $\boldsymbol{x}_k$. Importantly, each Hessian on the right hand side is taken with respect to the *activity* of hidden units, not with respect to the network *parameters* as it is typically defined.

To summarize, equations (1) through (4) provide four different methods to quantify pairwise similarity of hidden units. $\boldsymbol{S}^{\text{cov}}$ and $\boldsymbol{S}^{\text{i-sens}}$ are **upstream**, while $\boldsymbol{S}^{\text{o-sens}}$ and $\boldsymbol{S}^{\text{hess}}$ are **downstream**. All four take values in $[0, \infty)$, however, it is not clear if the raw magnitudes matter, or only relative (normalized) magnitudes. Take $\boldsymbol{S}^{\text{cov}}$ for instance. Kornblith et al. argue that hidden-unit *covariance* is more meaningful than *correlation*, since the principal directions of variation both affect learning and have been shown to correlate with meaningful perceptual differences for human observers. On the other hand, we do not want our measure of modularity, which is a global property of an entire layer, to be skewed by a small number of highly variable units. Further, the community-detection algorithm we use for clustering below has been primarily developed and tested on *unweighted* and undirected graphs. For these reasons, we introduce an optional **normalized** version of each of the above four un-normalized similarity measures:

$$\hat{\boldsymbol{S}}_{ij} \equiv \frac{\boldsymbol{S}_{ij}}{\max \left( \sqrt{\boldsymbol{S}_{ii} \boldsymbol{S}_{jj}}, \epsilon \right)} , \tag{5}$$

where $\epsilon$ is a small positive value included for numerical stability. Whereas $\boldsymbol{S}_{ij} \in [0, \infty)$, the normalized values are restricted to $\hat{\boldsymbol{S}}_{ij} \in [0, 1]$. In total, this gives us eight methods to quantify pairwise similarity. These can be thought of as 2x2x2 product of methods, as shown in the color scheme in Figure 2: the **upstream** vs **downstream** axis, the **unnormalized** vs **normalized** axis, and the **covariance** vs **gradient** (i.e. sensitivity) axis. We group together both $\boldsymbol{S}^{\text{cov}}$ and $\boldsymbol{S}^{\text{hess}}$ under the term "covariance" because the Hessian is closely related to the covariance of gradient vectors of the loss across inputs.

## 3.2   Quantifying modularity by clustering

Decomposing a set into clusters that are maximally similar within clusters and maximally dissimilar across clusters is a well-studied problem in graph theory and network science. In particular, Girvan & Newman proposed a method that cuts a graph into its maximally *modular* subgraphs (Girvan & Newman, 2002; Newman & Girvan, 2004; Newman, 2006), and this tool has previously been used to study modular neural networks (Amer & Maul, 2019; Béna & Goodman, 2021).

We apply this tool from graph theory to our problem of detecting functional modules in neural networks by constructing an adjacency matrix $\boldsymbol{A}$ from the similarity matrix $\boldsymbol{S}$ by simply removing the diagonal (self-similarity):

$$\boldsymbol{A}_{ij} \equiv \begin{cases} \boldsymbol{S}_{ij} & \text{if } i \neq j \\ 0 & \text{otherwise} \end{cases} .$$

Given $\boldsymbol{A}$, we can simplify later notation by first constructing the normalized adjacency matrix, $\tilde{\boldsymbol{A}}$, whose elements all sum to one:

$$\tilde{\boldsymbol{A}} \equiv \frac{\boldsymbol{A}}{\sum_{ij} \boldsymbol{A}_{ij}} ,$$

or, more compactly, $\tilde{\boldsymbol{A}} \equiv \boldsymbol{A}/\mathbf{1}_n^\top \boldsymbol{A} \mathbf{1}_n$ where $\mathbf{1}_n$ is a column vector of length $n$ containing all ones. Let $\boldsymbol{P}$ be an $n \times c$ matrix that represents cluster assignments for each of $n$ units to a maximum of $c$ different clusters. Cluster assignments can be "hard" ($\boldsymbol{P}_{ij} \in \{0, 1\}$) or "soft" ($\boldsymbol{P}_{ij} \in [0, 1]$), but in either case the constraint $\boldsymbol{P} \mathbf{1}_c = \mathbf{1}_n$ must be met, i.e. that the sum of cluster assignments for each unit is 1.[3] If an entire column of $\boldsymbol{P}$ is zero, that cluster is unused, so $c$ only provides an upper-limit to the number of clusters, and in practice we set $c = n$. Girvan & Newman propose the following score to quantify the level of "modularity" when partitioning the normalized adjacency matrix $\tilde{\boldsymbol{A}}$ into the cluster assignments $\boldsymbol{P}$:

$$Q(\tilde{\boldsymbol{A}}, \boldsymbol{P}) \equiv \mathrm{Tr}(\boldsymbol{P}^\top \tilde{\boldsymbol{A}} \boldsymbol{P}) - \mathrm{Tr}(\boldsymbol{P}^\top \tilde{\boldsymbol{A}} \mathbf{1}_n \mathbf{1}_n^\top \tilde{\boldsymbol{A}} \boldsymbol{P}) \,. \tag{6}$$

The first term sums the total connectivity (or, in our case, similarity) of units that share a cluster. By itself, this term is maximized when $\boldsymbol{P}$ assigns all units to a single cluster. The second term gives the *expected* connectivity within each cluster under a null model where the elements of $\tilde{\boldsymbol{A}}$ are interpreted as the joint probability of a connection, and so $\tilde{\boldsymbol{A}} \mathbf{1}_n \mathbf{1}_n^\top \tilde{\boldsymbol{A}}$ is the product of marginal probabilities of each unit's connections. This second term encourages $\boldsymbol{P}$ to place units into the same cluster only if they are more similar to each other than "chance." Together, equation (6) is maximized by partitioning $\tilde{\boldsymbol{A}}$ into clusters that are strongly intra-connected and weakly inter-connected.

We define the modularity of a set of neural network units as the maximum achievable $Q$ over all $\boldsymbol{P}$:

$$\begin{aligned} \boldsymbol{P}^*(\tilde{\boldsymbol{A}}) &\equiv \arg\max_{\boldsymbol{P}} Q(\tilde{\boldsymbol{A}}, \boldsymbol{P}) \\ Q^*(\tilde{\boldsymbol{A}}) &\equiv Q(\tilde{\boldsymbol{A}}, \boldsymbol{P}^*) \,. \end{aligned} \tag{7}$$

To summarize, to divide a given pairwise similarity matrix $\boldsymbol{S}$ into modules, we first construct $\tilde{\boldsymbol{A}}$ from $\boldsymbol{S}$, then we find the cluster assignments $\boldsymbol{P}^*$ that give the maximal value $Q^*$. Importantly, this optimization process provides two pieces of information: a *modularity score* $Q^*$ which quantifies the amount of modularity in a set of neurons, for a given similarity measure. We also get the actual *cluster assignments* $\boldsymbol{P}^*$, which provide additional information and can be compared across different similarity measures. Given a set of cluster assignments $\boldsymbol{P}^*$, we quantify the number of clusters by first getting the fraction of units in each cluster, $\boldsymbol{r}(\boldsymbol{P}^*) = \frac{\mathbf{1}_n^\top \boldsymbol{P}^*}{n}$. We then use the formula for discrete entropy to measure the dispersion of cluster sizes: $H(\boldsymbol{r}) = -\sum_{i=1}^c \boldsymbol{r}_i \log \boldsymbol{r}_i$. Finally we say that the number of clusters in $\boldsymbol{P}^*$ is

$$\text{num clusters}(\boldsymbol{P}^*) = e^{H(\boldsymbol{r}(\boldsymbol{P}^*))} \,. \tag{8}$$

We emphasize that discovering the number of clusters in $\boldsymbol{P}^*$ is included automatically in the optimization process; we set the maximum number of clusters $c$ equal to the number of hidden units $n$, but in our experiments we find that $\boldsymbol{P}^*$ rarely uses more than 6 clusters for hidden layers with 64 units (Supplemental Figure S7).

It is important to recognize that the sense of the word "modularity" in graph theory is in some important ways distinct from its meaning in terms of engineering functionally modular systems. Whether or not a high value of $Q*$ indicates more "modular" design requires additional interpretation beyond the choice of pairwise similarity measure $\boldsymbol{S}$. In graph-theoretic terms, a "module" is a cluster of nodes that are highly intra-connected and weakly inter-connected to other parts of the network, defined formally by $Q$ (Girvan & Newman, 2002; Newman & Girvan, 2004). This definition of graph modularity uses a particular idea of a "null model" based on random connectivity between nodes in a graph (Amer & Maul, 2019). While this null-model of graph connectivity enjoys a good deal of historical precedence in the theory of randomly-connected graphs, where unweighted graphs are commonly studied in terms of the probability of connection between random pairs of nodes, it is not obvious that the same sort of null model applies to groups of "functionally similar" units in an ANN. This relates to the earlier discussion of ISA, and provides a possibly unsatisfying answer to the question of what counts as a "surprising" amount of statistical independence between clusters; using $Q$ makes the implicit choice that the product of average pairwise similarity, $\tilde{\boldsymbol{A}} \mathbf{1}_n \mathbf{1}_n^T \tilde{\boldsymbol{A}}$, gives the "expected"

---

[3]While we only consider hard cluster assignments in our experiments below, we have encountered cases where $Q$ is maximized by soft cluster assignments, so we include it in the definition here for completeness.

similarity between units. An important problem for future work will be to closely reexamine the question of what makes neural populations *functionally* similar or dissimilar, above and beyond *statistical* similarity (Csordás et al., 2021; Kornblith et al., 2019), and what constitutes a surprising amount of (dis)similarity that may be indicative of modular design. Throughout this paper, our use of the term "modularity" to refer to $Q^*$ should be understood as only a quantitative proxy for truly modular design principles, but one that enjoys some precedence in the study of modular neural networks (Kashtan & Alon, 2005; Watanabe et al., 2018; Béna & Goodman, 2021).

Finding $P^*$ exactly is NP-complete (Brandes et al., 2008), so in practice we use a variation on the approximate method proposed by (Newman, 2006). Briefly, the approximation works in two steps: first, an initial set of cluster assignments is constructed using a fast spectral initialization method that, similar to other spectral clustering algorithms, recursively divides units into clusters based on the sign of eigenvectors of the matrix $\mathbf{B} = \tilde{A} - \tilde{A}\mathbf{1}_n\mathbf{1}_n^\top \tilde{A}$ and its submatrices. Only subdivisions that increase $Q$ are kept. In the second step, we use a Monte Carlo method that repeatedly selects a random unit $i$ then resamples its cluster assignment, holding the other $n - 1$ assignments fixed. This resampling step involves a kind of exploration/exploitation trade-off: $Q$ may decrease slightly on each move to potentially find a better global optimum. We found that it was beneficial to control the entropy of each step using a temperature parameter, to ensure that a good explore/exploit balance was struck for all $\tilde{A}$. Supplemental Figure S5 shows that both the initialization and the Monte Carlo steps play a crucial role in finding $P^*$, consistent with the observations of (Newman, 2006). Full algorithms are given in Appendix A.1.

## 4 Experiments

### 4.1 Setup and initial hypotheses

We would like to understand the behavior of the various notions of modularity above, operationalized using $Q^*$, for each of the eight different methods for quantifying pairwise similarity introduced in the previous section. In our initial experiments, we began by studying a large collection of simple feedforward fully-connected networks trained on the MNIST dataset (LeCun et al., 1998) across a range of regularization schemes (**Experiment 1**). In follow-up experiments, we further explored the effects of network width (**Experiment 2**) and depth (**Experiment 3**). A summary of the architectures and hyperparameters used in each experiment can be found in Supplemental Figure S1 and Table S1.

In all experiments, we defined $x$ (input layer) as the raw pixel inputs and $y$ (output layer) as the $10-$dimensional class logits. We discarded models that achieved less than $80\%$ test accuracy (this only happened at extreme values of regularization). Summary statistics of model behavior as a function of regularization strength are shown in Supplementary Figures S2 through S4. For reference values, we also performed all analyses on 100 randomly initialized models from each architecture with zero training. Models were written and trained using PyTorch (Paszke et al., 2019) and PyTorch Lightning[4], and all compute jobs were run on a private server and managed using GNU Parallel (Tange, 2011). Code is publicly available at `https://github.com/KordingLab/clustering-units-upstream-downstream`.

Before running these experiments, we hypothesized that

1. Dropout would decrease modularity by encouraging functions to be "spread out" over many units.

2. L2 regularization (weight decay) would minimally impact modularity, since the L2 norm is invariant to rotation while modularity depends on axis-alignment.

3. L1 regularization on weights would increase modularity by encouraging sparsity between subnetworks.

4. All similarity measures would be qualitatively consistent with each other.

As shown below, all four of these hypotheses turned out to be wrong, to varying degrees.

---

[4]https://www.pytorchlightning.ai/

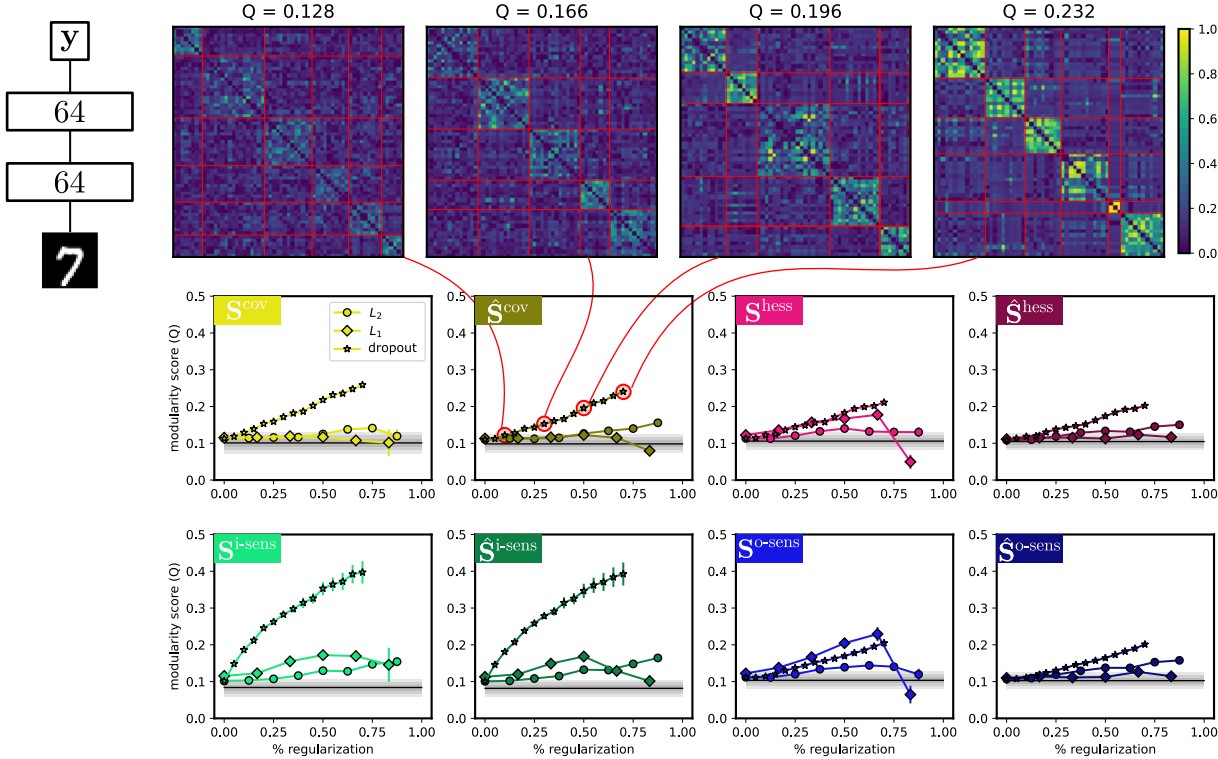

Figure 3: Modularity score as a function of regularization strength. **Inset network diagram:** Results for Experiment 1 are based on a simple feedforward network, trained on MNIST, with two hidden layers and 64 units per layer. **Top row:** example similarity matrices, sorted by cluster, and their associated $Q^*$ values. Thin red grid lines indicate boundaries between clusters discovered by the modularity-maximizing cluster assignments $\boldsymbol{P}^*$. Similarity in these examples is measured by $\hat{\boldsymbol{S}}^{\text{cov}}$, i.e. the absolute value correlation between hidden units' activity on test items, across increasing values of dropout regularization (curved red lines). **Bottom two rows:** modularity score ($Q^*$) as a function of percent regularization, combined for both hidden layers. Each subplot shows a different one of our eight similarity measures. Errorbars indicate standard error of the mean (across seeds and layers). Gray shading shows the distribution of scores for untrained models, out to $\pm 3\sigma$. Note that the x-axis, "percent regularization," is different for each series: it is identical to dropout probability for the dropout series, but it is log-spaced from $0\% = 1e-5$ to $100\% = 1e-1$ for $L_2$ and log-spaced from $0\% = 1e-5$ to $100\% = 1e-2$ for $L_1$.

## 4.2 How modularity depends on regularization (Experiment 1)

In Experiment 1, we trained a fully-connected neural network on MNIST with two hidden layers and 64 units per layer. Figure 3 shows the dependence of trained networks' modularity score ($Q^*$) as a function of regularization strength for each of three types of regularization: an L2 penalty on the weights (weight decay), an L1 penalty on the weights, and dropout. The top row of Figure 3 shows four example $\tilde{\boldsymbol{A}}$ matrices sorted by cluster, to help give an intuition behind the quantitative values of $Q^*$. In these examples, the increasing value of $Q^*$ is driven by an increasing contrast between intra-cluster similarity and inter-cluster similarity. In this example, it also appears that the *number* and *size* of clusters remains roughly constant; this observation is confirmed by plotting the number of clusters versus regularization strength in Supplemental Figure S7.

Figure 3 shows a number of surprising patterns that contradict our initial predictions. First, and most saliently, we had predicted that dropout would reduce modularity, but found instead that it has the *greatest* effect on $Q^*$ among the three regularization methods we tried. This is especially apparent in the upstream methods (first two columns of the figure), and is also stronger for the first hidden layer than the second

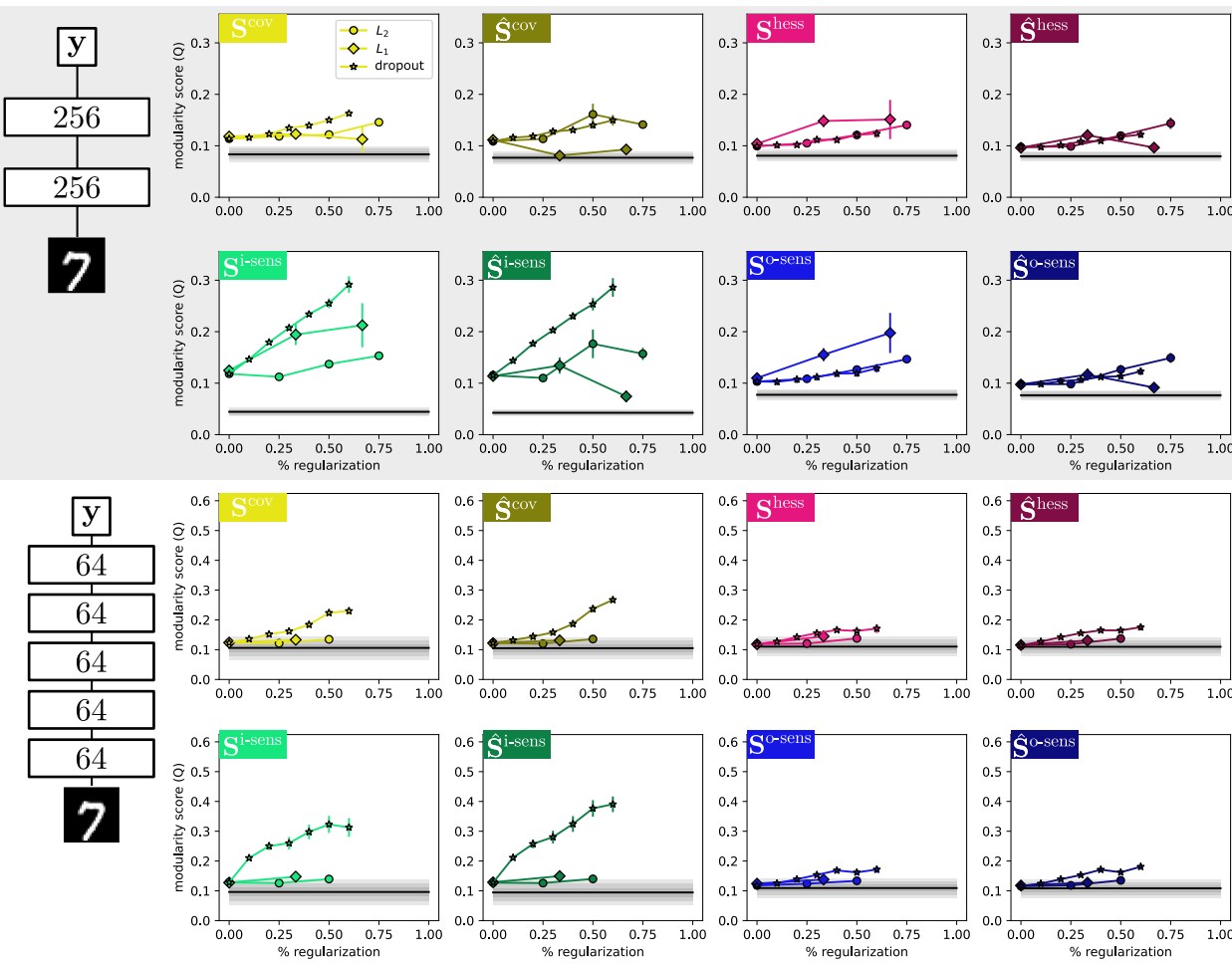

Figure 4: Modularity score as a function of regularization strength for wider (gray background, top) and deeper (white background, bottom) models. Subplots are formatted identically to Figure 3.

(Supplemental Figure S6). In general, $Q^*$ can increase either if the network partitions into a greater number of clusters, or if the contrast between clusters is exaggerated. We found that this dramatic effect of dropout on $Q^*$ was accompanied by only minor changes to the number of clusters (Supplemental Figure S7), and so we can conclude that **dropout increases $Q^*$ by increasing the redundancy of hidden units**. In other words, hidden units become more clusterable because they are driven towards behaving like functional replicas of each other, separately for each cluster. This observation echoes, and may explain, why dropout also increases the "clusterability" of network weights in a separate study (Filan et al., 2021).

The second surprising result in Figure 3 is that L2 regularization on the weights did, in fact, increase $Q^*$, whereas we had expected it to have no impact. Third, L1 regularization had a surprisingly weak effect, although its similarity to the L2 regularization results may be explained by the fact that they actually resulted in fairly commensurate sparsity in the trained weights (Supplemental Figure S2 bottom row). Fourth, we had expected few differences between the eight different methods for computing similarity, but there appear to be distinctive trends by similarity type both in Figure 3 as well as in the number of clusters detected (Supplemental Figure S7). The next section explores the question of similarity in the results in more detail.

### 4.3 Generalization to wider and deeper networks (Experiments 2 and 3)

We repeated this analysis of $Q*$ scores and clusters, as a function of regularization strength, for both a wider and deeper version of the fully-connected network used in Experiment 1. Modularity scores for these further experiments are shown in Figure 4, and analogous plots showing their numbers of clusters can be found in Supplemental Figure S8.

While many of the qualitative trends remain, two differences to Experiment 1 are worth noting. First, inspecting the results for wider models, although we again see that dropout increases $Q*$, its effect relative to the other normalization types is greatly attenuated. In fact, in this wider architecture, we begin to see some evidence for our original hypothesis that increasing L1 weight regularization would increase modularity, at least with respect to downstream measures of similarity. However, this is a fragile effect, as too-large L1 regularization quickly overpowers learning and degrades performance (Supplemental Figure S3). Second, the deeper models in Experiment 3 show an even more exaggerated difference between dropout and other regularization types, at least for upstream similarity measures. We further found that this effect is primarily driven by layers early in the network (see Supplemental Figures S9 and S10 for a breakdown by layer).

Another interesting effect revealed by Experiment 3 is that the effect of dropout on $Q^*$ was highest in early layers for *upstream* similarity measures, but highest in later layers for *downstream* similarity measures (Supplemental Figure S9). As will be explored in more detail in the next section, this provides some initial evidence that the upstream- and downstream-families of approaches are measuring different things.

### 4.4 Comparing modules discovered by different similarity methods

The previous two sections discussed idiosyncratic trends in the modularity scores $Q^*$ as a function of both regularization strength and how pairwise similarity between units ($\boldsymbol{S}$) is computed. However, such differences in the quantitative value of $Q^*$ are difficult to interpret, and would largely be moot if the various methods agreed on the question of *which units belong in which cluster*. We now turn to the question of how similar the cluster assignments $\boldsymbol{P}^*$ are across our eight definitions of functional modules. To minimize ambiguity, we will use the term "functional-similarity" to refer to $\boldsymbol{S}$, and "cluster-similarity" to refer to the comparison of different cluster assignments $\boldsymbol{P}^*$ across different functional-similarity methods. Note that all comparisons between cluster assignments investigated in this section are performed separately per layer per model – cluster assignments are never compared for units across different models.

Quantifying similarity between cluster assignments is a well-studied problem, and we tested a variety of methods in the `clusim` Python package (Correia et al., 2018). All cluster-similarity methods we investigated gave qualitatively similar results, so here we report only the "Element Similarity" method of Gates et al., which is a value between 0 and 1 that is small when two cluster assignments are unrelated, and large when one cluster assignment is highly predictive of the other. However, it is conceivable that two cluster assignments do not strongly predict each other, but nonetheless they give rise to similar values of $Q$. That is, to the extent that $Q$ has many good maxima as a function of $\boldsymbol{P}^*$, differences between cluster assignments is not indicative of strong differences in the similarity structures.[5] For this reason, we included an additional "transfer" test of cluster alignment: let $\boldsymbol{P}_A^*$ be the optimal cluster assignments computed for the pairwise similarities $\tilde{\boldsymbol{A}}_A$, and likewise $\boldsymbol{P}_B^*$ for $\tilde{\boldsymbol{A}}_B$. Then,

$$\frac{Q(\tilde{\boldsymbol{A}}_A, \boldsymbol{P}_B^*) + Q(\tilde{\boldsymbol{A}}_B, \boldsymbol{P}_A^*)}{2} \tag{9}$$

quantifies how well the cluster assignments $\boldsymbol{P}^*$ transfer between similarity measures, transferring both B to A and vice versa. We refer to (9) as the "transferability" of clusters between two similarity measures.

Note that all of these cluster-similarity analyses are applied only to $\boldsymbol{P}^*$ cluster assignments computed *in the same layer of the same model*. Thus, any *dissimilarity* in clusters that we see is due entirely to the different choices for functional-similarity, $\boldsymbol{S}$, rather than to differences in regularization, seed, architecture, etc.

Figure 5a summarizes the results of this cluster-similarity analysis in Experiment 1: **there is a striking difference between clusters of units identified by "upstream" functional-similarity methods**

---

[5]We are indebted to an anonymous reviewer for pointing this out.

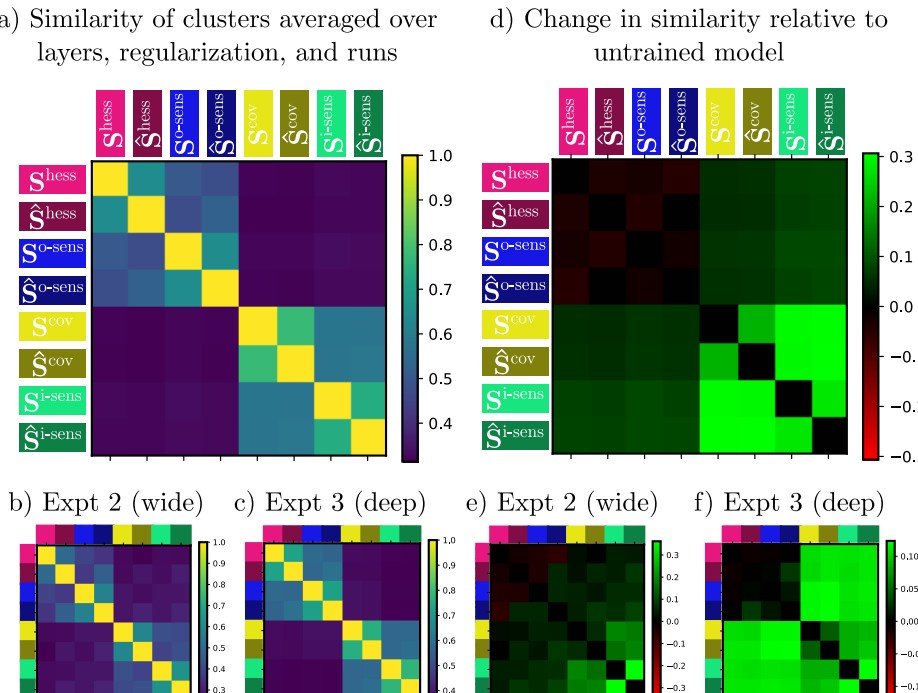

Figure 5: Summary of cluster-similarity analysis. **a)** Separately for each model, layer, and run, we computed $\binom{8}{2}$ cluster-similarity scores between cluster assignments $\boldsymbol{P}^*$ for each pair of the eight functional-similarity methods ($\boldsymbol{S}$) above. This plot shows the *average* cluster-similarity for each pair of functional-similarity methods in Experiment 1. **b-c)** Same as (a) but for wider (Experiment 2) and deeper (Experiment 3) models. **d)** We repeated the analysis in (a) on a set of 100 random (untrained) models. This panel shows the difference between (a) and the random models. Where the difference is positive (green), training had the effect of *increasing* cluster-similarity, and where it is zero (black), there is little difference in cluster-similarity before and after training. **e-f)** Same as (d) but for wider (Experiment 2) and deeper (Experiment 3) models. In all cases, the untrained reference models were of the same architecture as the trained model being analyzed. Note the different colorbar scale in panel (f).

($S^{\text{cov}}$, $\hat{S}^{\text{cov}}$, $S^{\text{i-sens}}$, $\hat{S}^{\text{i-sens}}$) **compared to "downstream" functional-similarity methods** ($S^{\text{hess}}$, $\hat{S}^{\text{hess}}$, $S^{\text{o-sens}}$, $\hat{S}^{\text{o-sens}}$). This analysis also reveals secondary structure within each class of upstream and downstream methods, where the choice to normalize not ($S$ vs $\hat{S}$) appears to matter little, and where there is a moderate difference between moment-based methods ($S^{\text{cov}}$, $S^{\text{hess}}$) and gradient-based methods ($S^{\text{i-sens}}$, $S^{\text{o-sens}}$). It is worth noting that some of this secondary structure is not robust across all types and levels of regularization; in particular, increasing L2 or L1 regularization strength appears to lead to (i) stronger dependence on normalization in the downstream methods, and (ii) a stronger overall agreement among the upstream methods (Supplemental Figure S11). We found that a qualitatively similar division between clusters identified using upstream and downstream similarity methods persisted in Experiments 2 and 3 (Figure 5b-c).

We next asked to what extent these cluster-similarity results are driven by training. As shown in Figure 5d-f, much of the structure in the downstream methods is unaffected by training (i.e. it is present in untrained models as well), while the cluster-similarity among different upstream methods only emerged as a result of training. Interestingly, despite the seemingly sharp differences between upstream and downstream methods shown in Figure 5a-c, this analysis further shows that training tended to weakly increase the agreement between upstream and downstream methods.

Overall, these trends in alignment between clusters were confirmed in the "transfer" analysis defined in equation (9) (Supplemental Figure S12).

## 5 Conclusions

An abundance of "modular" designs in engineered and evolved systems has led many to speculate about the usefulness of modularity as a design principle (Lipson, 2007), and how modular designs might be discovered by learning agents such as artificial neural networks (Amer & Maul, 2019). Yet, precisely defining what a "module" is in a neural network is an open problem. Here we operationalized modules in a neural network as *clusters of hidden units that perform similar functions*, which leads naturally to the related question of *what makes any given pair of units functionally similar?* We introduced eight functional-similarity measures intended to cover a variety of intuitions about what makes two units similar, and empirically evaluated cluster-assignments based on each of these eight methods in a large number of trained models. It should be emphasized that our quantitative results on increased or decreased "modularity" refer to changes in $Q^*$, which may not correspond to the kind of system-level "modularity" that motivates work in this area. Our goal is not to establish an indisputable definition of modularity, but to call attention to surprising ways that intuitions about modularity can be wrong.

One such surprising observation was that dropout increases modularity (as defined by $Q^*$) (Filan et al., 2021), although this has little to do with the common-sense definition of a "module." Instead, it is the byproduct of dropout causing subsets of units to behave like near-copies of each other, perhaps so that if one unit is dropped out, a copy of it provides similar information to the subsequent layer. This effect was reduced, but not absent, in the "wide" model of Experiment 2. To our knowledge, this redundancy-inducing effect of dropout has not been noted in the literature previously.

Our main result is that there is a crucial difference between defining "function" in terms of how units are driven by upstream inputs, and how units drive downstream outputs. While we studied this distinction between upstream and downstream similarity in the context of modularity and clustering, it speaks to the deeper and more general problem of how best to interpret neural representations. For example, some sub-disciplines of representation-learning (e.g. "disentanglement") have long emphasized that a "good" neural representation is one where distinct features of the world drive distinct sub-populations or sub-spaces of neural activity (Higgins et al., 2018; Eastwood & Williams, 2018; Ridgeway & Mozer, 2018). This is an upstream way of thinking about what is represented, since it depends only on the relationship between inputs and the unit activations and does not take into account what happens downstream. Meanwhile, many have argued that the defining characteristic of a neural representation is its causal role in downstream behavior (Garson & Papineau, 2019); this is, of course, a downstream way of thinking. At a high level, one way to interpret our results is is that **upstream and downstream ways of thinking about neural representations are not necessarily aligned, even in trained networks**. This observation is reminiscent of recent empirical

work finding that "disentangled" representations in auto-encoders (an upstream concept) do not necessarily lead to improved performance or generalization to novel tasks (a downstream concept) (Locatello et al., 2019; Montero et al., 2021) (cf. (van Steenkiste et al., 2019)).

Despite its theoretical motivations, this is an empirical study. We trained over 300 feedforward, fully-connected neural networks on MNIST. While it is not obvious whether MNIST admits a meaningful "modular" solution, Experiments 2 and 3 suggest that our main result – the misalignment between upstream and downstream definitions of neural similarity – is fairly robust to width and depth. Still, it is plausible that upstream and downstream ways of thinking about neural representations will become better aligned in more structured tasks and larger models.

Our work raises the important questions: are neural representations defined by their inputs or their outputs? And, in what contexts is it beneficial for these to be aligned? We look forward to future work applying our methods to larger networks trained on more structured data, as well as recurrent networks. We also believe it will be valuable to evaluate the effect of attempting to maximize modularity, as we have defined it, during training, to see to what extent this is possible and whether it leads to performance benefits. Note that maximizing $Q$ during training is challenging because (i) computing $S$ may require large batches, and more importantly (ii) optimizing $Q$ is highly prone to local minima, since neural activity and cluster assignments $P$ will tend to reinforce each other, entrenching accidental clusters that appear at the beginning of training. We suspect that maintaining uncertainty over cluster assignments (e.g. using soft $P_{ij} \in [0,1]$ rather than hard $P \in \{0,1\}$ cluster assignments) will be crucial if optimizing any of our proposed modularity metrics during training.

Unequivocally defining and quantifying the modularity of a neural network remains an open problem. However, we hope that the distinction we introduced between upstream and downstream ways of thinking about modules, and about neural representations more generally, will foster and sharpen future work on modular neural network design.

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

# A  Appendix

## A.1  Algorithms

This section gives pseudocode for the algorithm we used to compute clusters $\boldsymbol{P}^*$ from the normalized matrix of pairwise associations between units, $\tilde{\boldsymbol{A}}$. Before running these algorithms, we always remove all-zero rows and columns from $\tilde{\boldsymbol{A}}$; we consider these units to all be in a separate "unused" cluster.

---

**Algorithm 1** Full clustering algorithm.

---

**Require:** Normalized pairwise associations $\tilde{\boldsymbol{A}}$
1: $\boldsymbol{P} \leftarrow \text{GREEDYSPECTRALMODULES}(\tilde{\boldsymbol{A}})$         ▷ Initialize $\boldsymbol{P}$ using spectral method
2: $\boldsymbol{P}^* \leftarrow \text{MONTECARLOMODULES}(\tilde{\boldsymbol{A}}, \boldsymbol{P})$     ▷ Further refine $\boldsymbol{P}$ using Monte Carlo method
3: **return** $\boldsymbol{P}^*$

---

---

**Algorithm 2** Pseudocode for greedy, approximate, spectral method for finding modules

---

1: **function** GREEDYSPECTRALMODULES($\tilde{\boldsymbol{A}}$)
2:   $\boldsymbol{B} \leftarrow \tilde{\boldsymbol{A}} - \tilde{\boldsymbol{A}}\mathbf{1}_n\mathbf{1}_n^\top\tilde{\boldsymbol{A}}$   ▷ $\boldsymbol{B}$ is analogous to the graph Laplacian, but for modules (Newman, 2006)
3:   $\boldsymbol{P} \leftarrow \begin{bmatrix}\mathbf{1}\,\mathbf{0}\,\mathbf{0}\dots\mathbf{0}\end{bmatrix}$    ▷ Initialize $\boldsymbol{P}$ to a single cluster, which will be (recursively) split in two
4:   queue $\leftarrow [0]$       ▷ FILO queue keeping track of which cluster we'll try splitting next
5:   $Q \leftarrow \text{Tr}(\boldsymbol{P}^\top\boldsymbol{B}\boldsymbol{P})$             ▷ Compute $Q$ for the initial $\boldsymbol{P}$
6:   **while** queue is not empty **do**
7:    $c \leftarrow \text{queue.pop()}$           ▷ Pop the next (leftmost) cluster id
8:    $i \leftarrow$ indices of all units currently in cluster $c$ according to $\boldsymbol{P}$
9:    $\boldsymbol{v} \leftarrow \text{eig}(\boldsymbol{B}(i,i))$   ▷ Get the leading eigenvector of the submatrix of $\boldsymbol{B}$ containing just units in $c$
10:    $i^+ \leftarrow$ subset of $i$ where $\boldsymbol{v}$ was positive     ▷ Split $\boldsymbol{v}$ by sign (if not possible, continue loop)
11:    $i^- \leftarrow$ subset of $i$ where $\boldsymbol{v}$ was negative
12:    $c' \leftarrow$ index of the next available (all zero) column of $\boldsymbol{P}$
13:    $\boldsymbol{P}' \leftarrow \boldsymbol{P}$ but with all $i^-$ units moved to cluster $c'$   ▷ Try splitting $c$ into $c$, $c'$ based on sign of $\boldsymbol{v}$
14:    $Q' \leftarrow \text{Tr}(\boldsymbol{P}'^\top\boldsymbol{B}\boldsymbol{P}')$        ▷ Compute updated $Q$ for newly-split clusters $\boldsymbol{P}'$
15:    **if** $Q' > Q$ **then**               ▷ Did splitting $c$ help?
16:     $Q, \boldsymbol{P} \leftarrow Q', \boldsymbol{P}'$             ▷ Update $Q$ and $\boldsymbol{P}$
17:     queue.append($c, c'$)    ▷ Push $c$ and $c'$ onto the queue to consider further subdividing them later.
18:    **else**
19:     ▷ Nothing to do - splitting $c$ into $c'$ did not improve $Q$, so we don't add further subdivisions to the queue, and we keep the old $\boldsymbol{P}$, $Q$ values
20:    **end if**
21:   **end while**
22:   **return** $\boldsymbol{P}$     ▷ Once the queue is empty, $\boldsymbol{P}$ contains a good initial set of cluster assignments
23: **end function**

---

---

**Algorithm 3** Pseudocode for Monte Carlo method for improving clusters.

---

**function** MONTECARLOMODULES($\tilde{A}, P, n$)
    **for** $n$ steps **do**
        $i \leftarrow$ index of a single a randomly selected unit
        $c \leftarrow$ index of the first empty cluster in $P$
        $Q^*, P^* \leftarrow \text{Tr}(P^\top(\tilde{A} - \tilde{A}\mathbf{1}_n\mathbf{1}_n^\top\tilde{A})P), P$         ▷ Keep track of best $Q, P$ pair found so far
        **for** $j = 1 \dots c$ **do**         ▷ Try moving unit $i$ to each cluster $j$, including a new cluster at $c$
            $P' \leftarrow P$ with $i$ reassigned to cluster $j$
            $Q'_j \leftarrow \text{Tr}(P'^\top(\tilde{A} - \tilde{A}\mathbf{1}_n\mathbf{1}_n^\top\tilde{A})P')$         ▷ Compute updated $Q$ with re-assigned unit
            **if** $Q'_j > Q^*$ **then**
                $Q^*, P^* \leftarrow Q'_j, P'$         ▷ Update $Q^*, P^*$ pair, even if we don't select this $j$ later
            **end if**
        **end for**
        $\tau \leftarrow$ whatever temperature makes p $\propto e^{Q'/\tau}$ have entropy $H = 0.15$
        p $\leftarrow \dfrac{e^{Q'/\tau}}{\sum_j e^{Q'_j/\tau}}$         ▷ We found $H = 0.15$ strikes a good balance between exploration and greedy
ascent.[6]
        $j^* \sim$ p         ▷ Sample new cluster assignment $j$ from categorical distribution p
        $P \leftarrow P$ with unit $i$ reassigned to cluster $j^*$, ensuring only the leftmost columns have nonzero values
    **end for**
    **return** $P^*$
**end function**

---

## A.2 Supplemental Figures

| Experiment (model) | L2 (weight decay) | L1 weight penalty | dropout prob. |
|---|---|---|---|
| Experiment 1 (MNIST $(64, 64)$) 9 seeds per hyperparameter | `logspace(-5,-1,9)` | 0.0 | 0.0 |
| | 1e-5 | `logspace(-5,-2,7)` | 0.0 |
| | 1e-5 | 0.0 | `linspace(0.05,0.7,14)` |
| Experiment 2 (MNIST $(256, 256)$) 3 seeds per hyperparameter | `logspace(-5,-1,5)` | 0.0 | 0.0 |
| | 1e-5 | `logspace(-5,-2,4)` | 0.0 |
| | 1e-5 | 0.0 | `linspace(0.1,0.6,6)` |
| Experiment 3 (MNIST $(64, 64, 64, 64, 64)$) 3 seeds per hyperparameter | `logspace(-5,-1,5)` | 0.0 | 0.0 |
| | 1e-5 | `logspace(-5,-2,4)` | 0.0 |
| | 1e-5 | 0.0 | `linspace(0.1,0.6,6)` |

Table S1: Models and hyperparameters. Number of units in each hidden layer given in parentheses in the first column, i.e. "MNIST $(64, 64)$" is a MLP with two hidden layers with 64 units in each layer. Each row of the table describes one hyperparameter sweep performed for the corresponding model. L2 regularization was always set to a minimum of $1e-5$ to avoid weights growing unboundedly (see Figures S2 through S4 for performance metrics and weight norms of trained models).

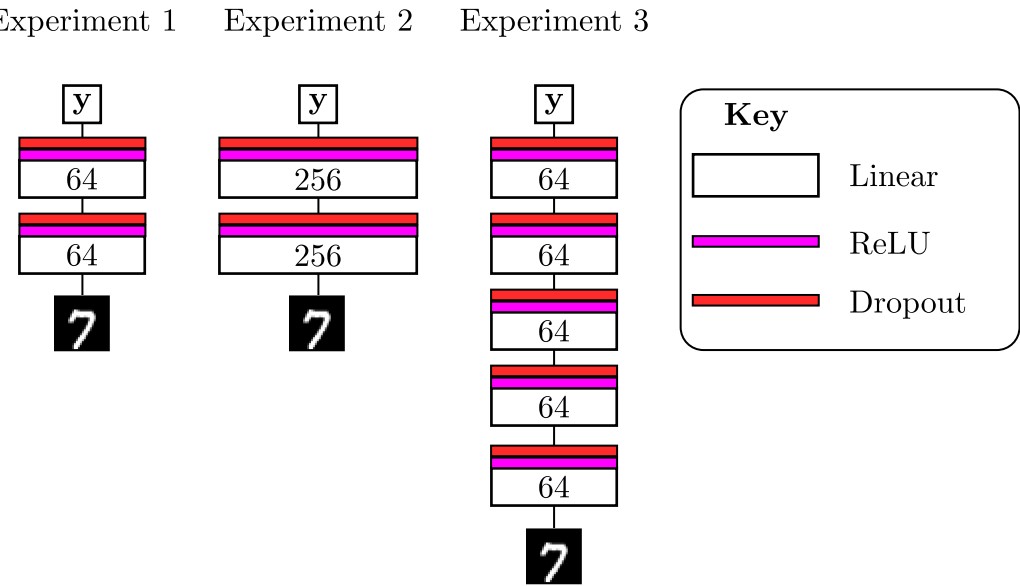

Figure S1: Graphical depiction of architectures used in Experiments 1 through 3.

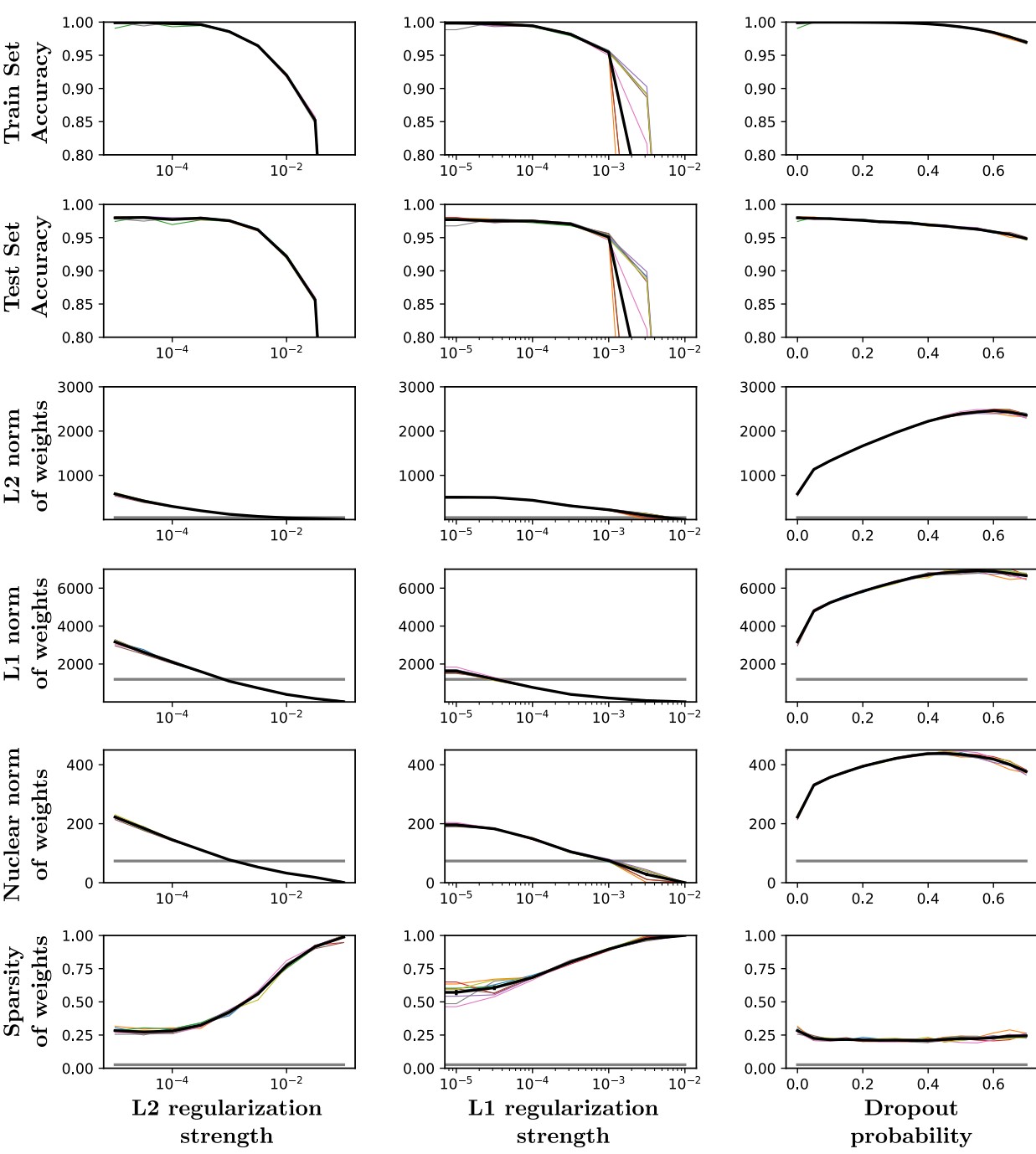

Figure S2: Basic performance metrics as a function of regularization strength for basic MNIST model (Experiment 1 in Table S1). Each column corresponds to a different regularization method, as in Table S1. Each row shows a metric calculated on the trained models. Thin colored lines are individual seeds, and thick black line is the average ± standard error across runs. Horizontal gray line shows each metric computed on randomly initialized network. Sparsity (bottom row) is calculated as the fraction of weights in the interval $[-1e-3, +1e-3]$.

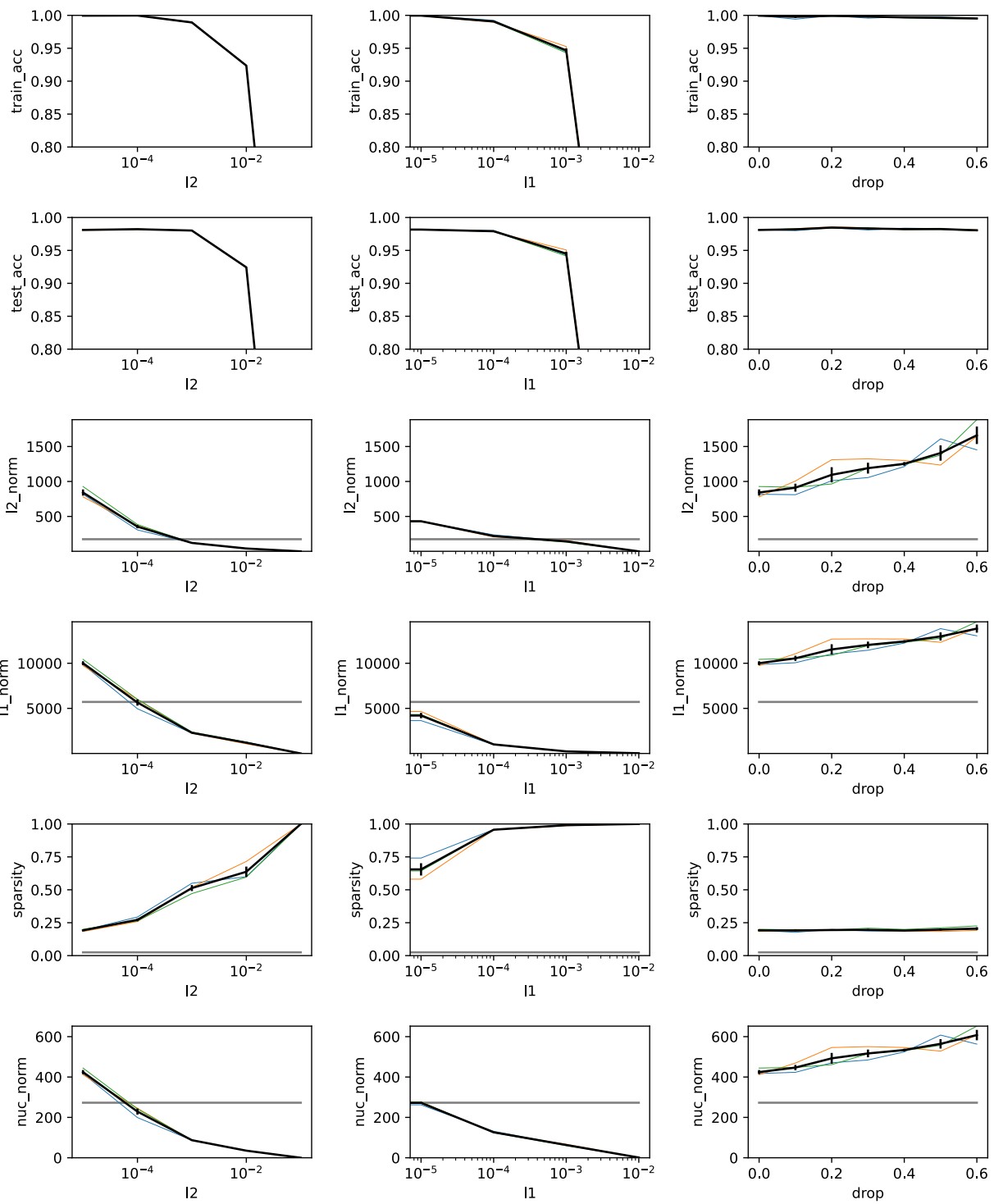

Figure S3: Basic performance metrics as a function of regularization strength for "wide" MNIST model (Experiment 2 in Table S1), plotted in the same format as Figure S2.

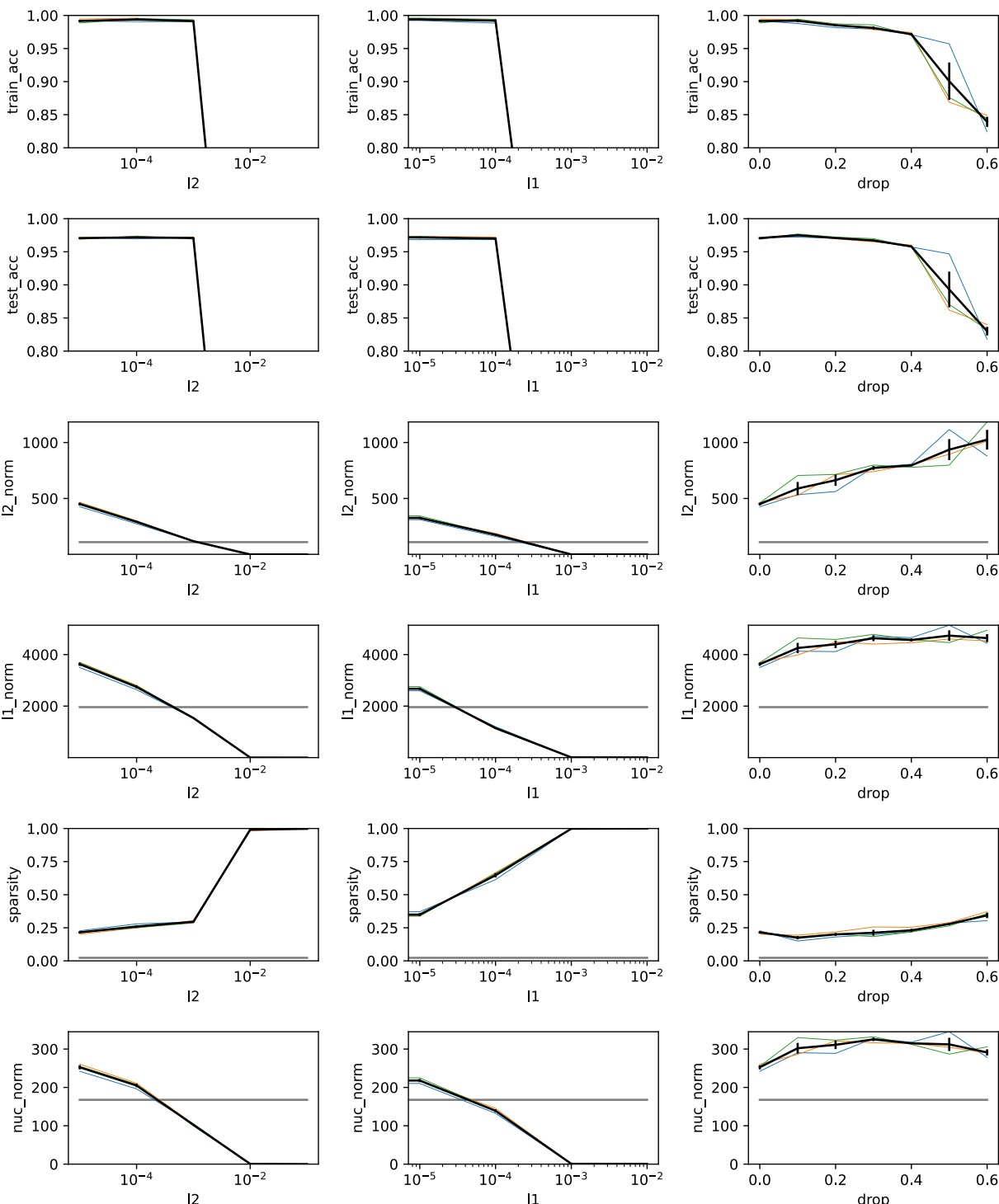

Figure S4: Basic performance metrics as a function of regularization strength for "deep" MNIST model (Experiment 3 in Table S1), plotted in the same format as Figure S2.

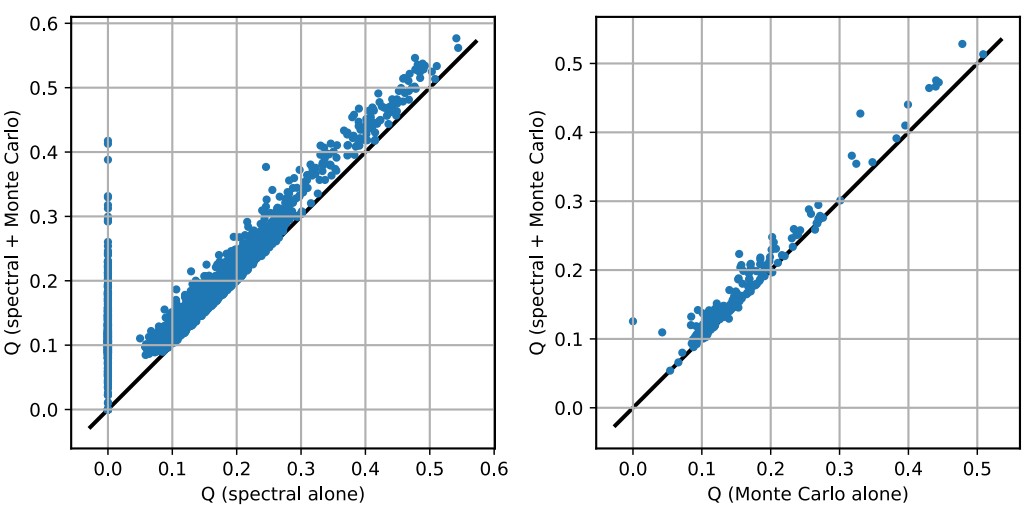

Figure S5: Both spectral initialization and Monte Carlo optimization steps contribute to finding a good value of $Q^*$. **Left:** The x-axis shows modularity scores ($Q^*$) achieved using only the greedy spectral method for finding $\boldsymbol{P}^*$. The y-axis shows the actual scores we used in the paper by combining the spectral method for initialization plus Monte Carlo search. The fact that all points are on or above the y=x line indicates that the Monte Carlo search step improved modularity scores. **Right:** The x-axis now shows modularity scores ($Q^*$) achieved using 1000 Monte Carlo steps, after initializing all units into a single cluster (we chose a random 5% of the similarity-matrices that were analyzed in the main paper to re-run for this analysis, which is why there are fewer points in this subplot than in the left subplot). The fact that all points are on or above the y=x line indicates that using the spectral method to initialize improved the search.

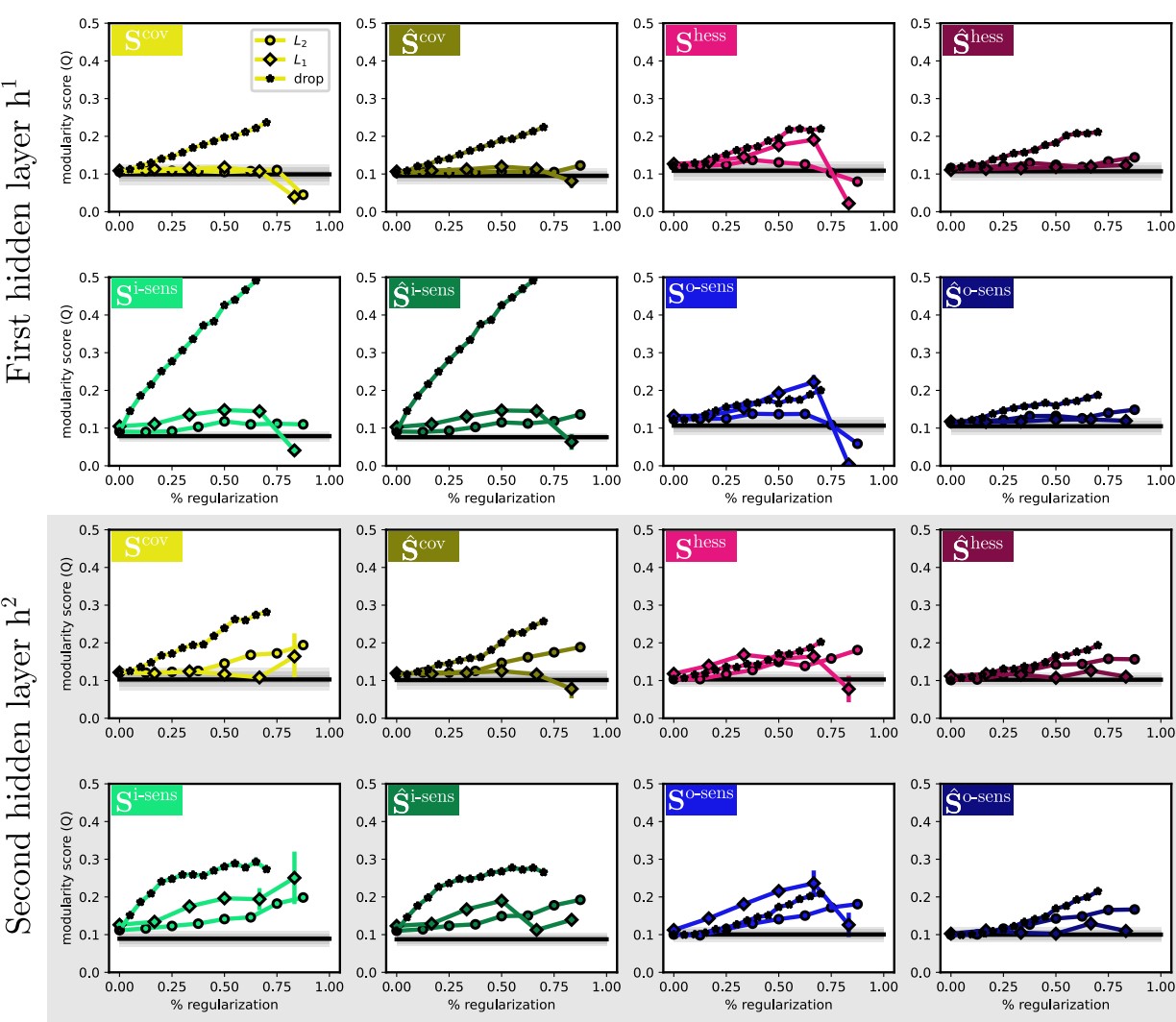

Figure S6: Modularity score ($Q^*$) versus regularization, split by layer. Format is identical to Figure 3, which shows modularity scores *averaged across layers*. Here, we break this down further by plotting each layer separately. The network used in our experiments has two hidden layers. The first two rows (white background) shows modularity scores for the first hidden layer $h^1$, and the last two rows (gray background) shows $h^2$.

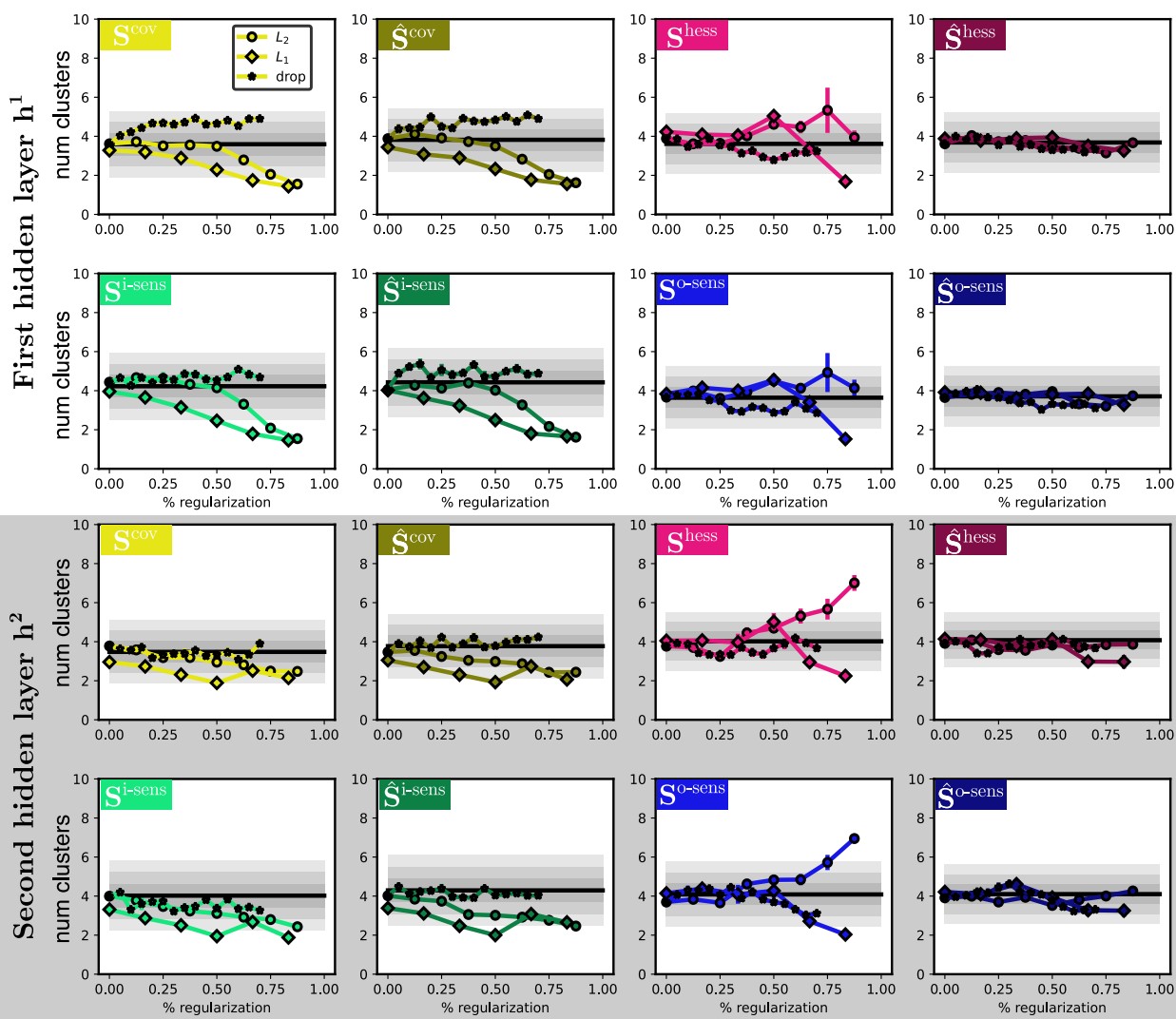

Figure S7: Number of clusters in $\boldsymbol{P}^*$ versus regularization, split by layer. Layout is identical to Figure S6. Gray shading in the background shows $1\sigma$, $2\sigma$, and $3\sigma$ quantiles of number of clusters in untrained (randomly initialized) networks. Note that, for the most part, training has little impact on the number of clusters detected, suggesting that consistently finding on the order of 2-6 clusters is more a property of the MNIST dataset itself than of training.

We computed the number of clusters using equation (8). This measure is sensitive to both the number and relative size of the clusters.

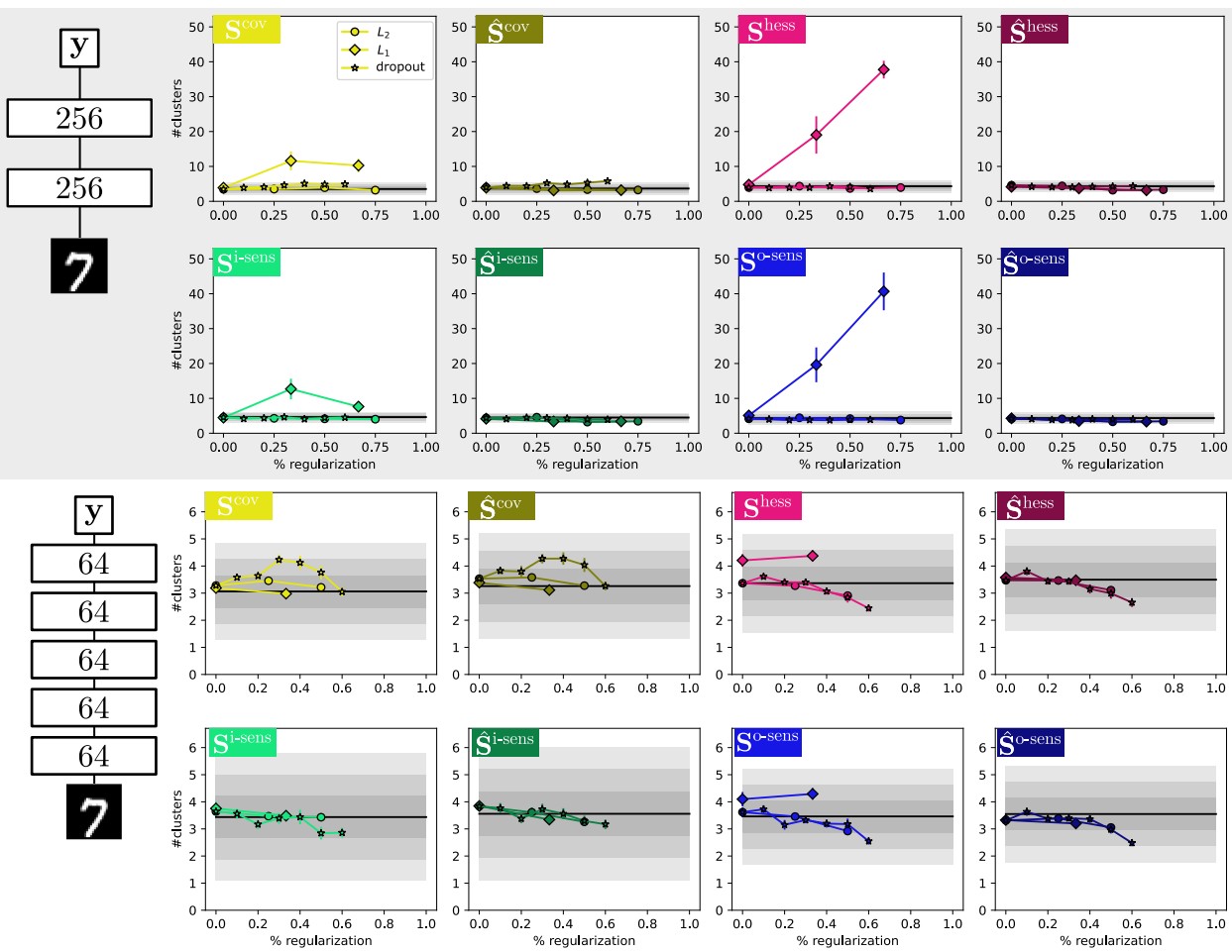

Figure S8: Number of clusters in $\boldsymbol{P}^*$ versus regularization, for wider (Experiment 2) and deeper (Experiment 3) models. Layout is identical to Figure S6.

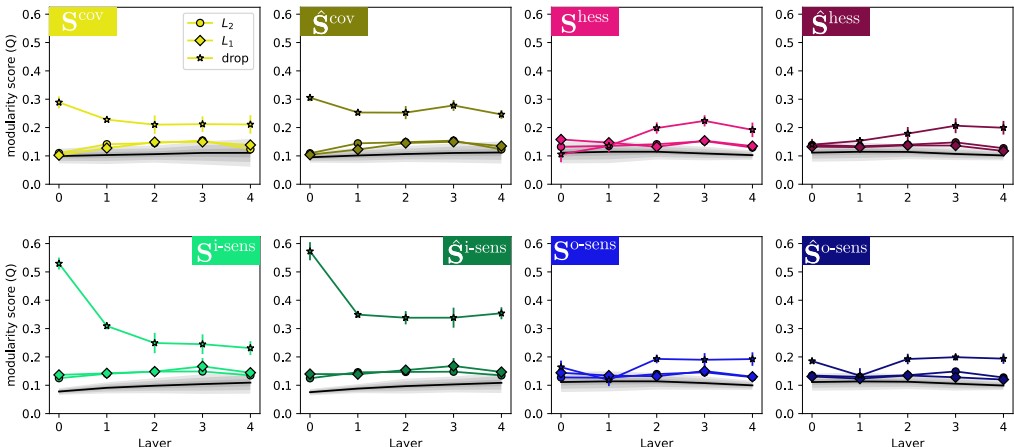

Figure S9: Modularity score versus depth in a 5-layer feedforward network (Experiment 3), using the maximum "good" regularization strength of each type, defined as largest value for each type of regularization that achieved at least 80% performance with at least half of the seeds. Gray shaded backgrounds show distribution of $Q^*$ values for untrained models; note that this null distribution itself depends on depth.

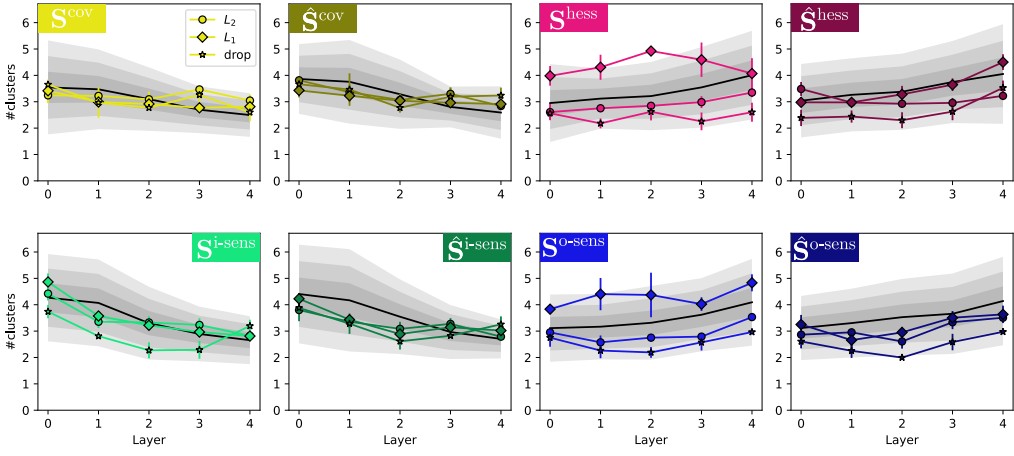

Figure S10: Number of clusters versus depth in a 5-layer feedforward network (Experiment 3), using the maximum "good" regularization strength of each type, defined as largest value for each type of regularization that achieved at least 80% performance with at least half of the seeds. Gray shaded backgrounds show distribution of $Q^*$ values for untrained models; note that this null distribution itself depends on depth.

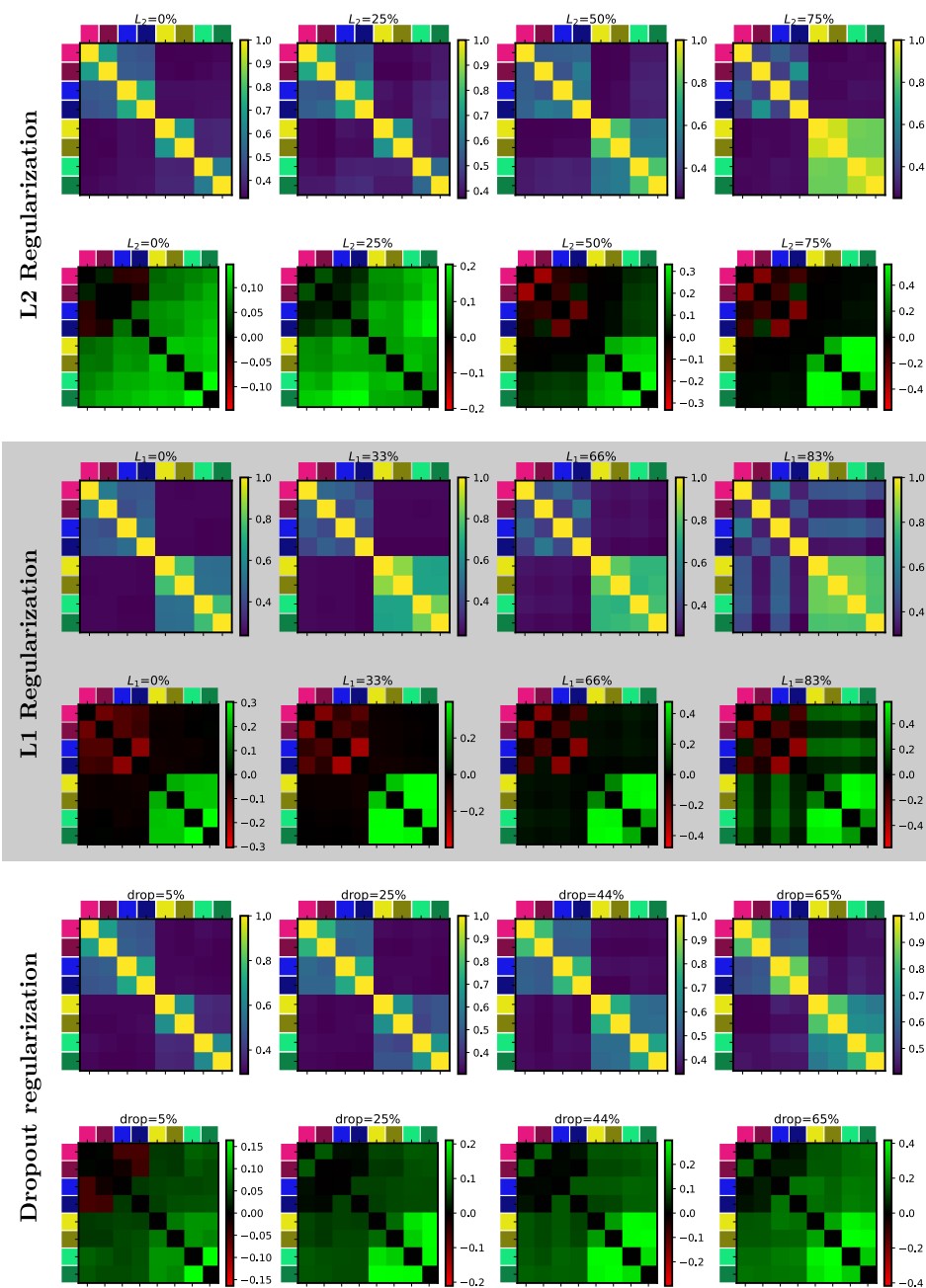

Figure S11: Further breakdown of cluster-similarity by regularization strength (increasing left to right) and type (L2/L1/dropout). Results in Figure 5 reflect an average of the results shown here. The six rows of this figure should be read in groups of two rows: in each group, the top row shows the similarity scores (averaged over layers and runs), and the bottom row shows the difference to untrained models. A number of features are noteworthy here: (i) at low values of all three types of regularization, there is little cluster-similarity within the upstream methods, but it becomes very strong at as regularization strength grows; (ii) at the highest values of L2 and L1 regularization, the pattern inside the 4x4 block of downstream methods changes to depend more strongly on normalization; (iii) a moderate amount of agreement between upstream and downstream methods is seen for large L1 regularization strength, but curiously only for *unnormalized* downstream methods.

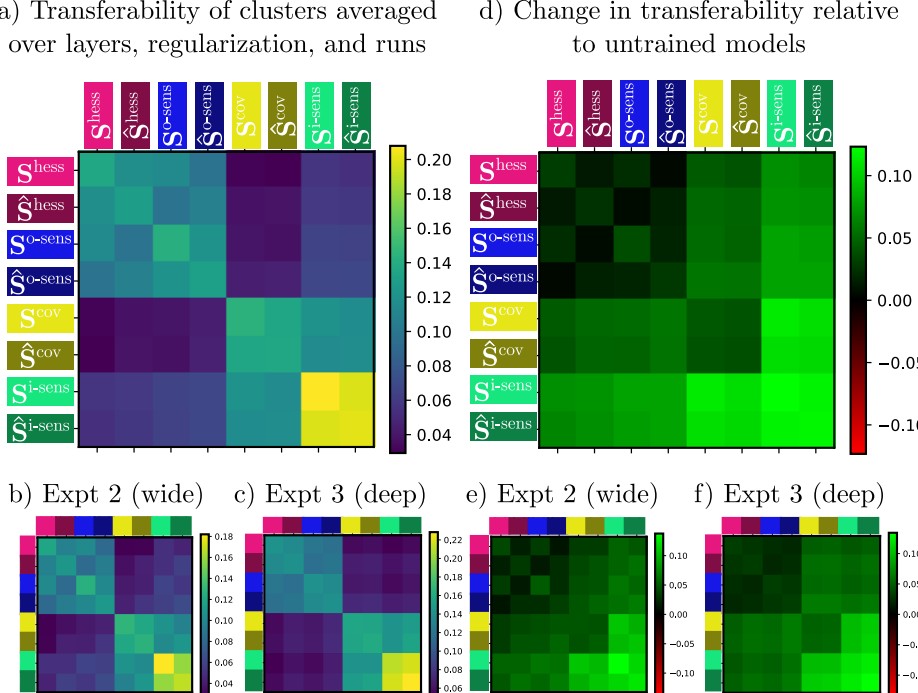

Figure S12: Cluster-transfer control for the cluster-alignment analysis, as in Figure 5 in the main text. "Transferability" is defined in equation (9). Note that transferability is not normalized, but is defined in raw units of $Q$. This is why the diagonals in panels (a)-(c) are not one.

