# OpenReview forum: "Clustering units in neural networks: upstream vs downstream information"
_TMLR — Accepted by TMLR_

### Review · Reviewer_g9XM · 2022-04-10

**Summary Of Contributions:**

This paper investigates a simple empirical question: how modular are neural network representations? That is, to what extent do trained artificial networks contain "clusters of similar-functioning units."

The authors propose several alternative similarity measures and use these in concert with simple graph theoretic methods to quantify modularity. A thoughtful review of prior work is provided. In my mind, the main advance seems to be that the authors used a broader set of hidden unit similarity measures than prior work, and that these different measures seem to lead one to different empirical conclusions in practice.

I did not find any technical errors. As I explain below I think the biggest weakness of the work is that there are rather limited empirical characterization of modularity across different architectures and different tasks.

**Broader Impact Concerns:**

None.

**Requested Changes:**

Though it is not critical to secure my recommendation, I would suggest that the authors add more experiments with (a) convolutional architectures, and (b) in more complex tasks (e.g. CIFAR-10 and CIFAR-100). This would increase the impact of this work by investigating the generalizability of their observations and conclusions. If further experiments will not be performed, I request that the authors add a few more sentences discussing this point at the end of their paper so that readers can take appropriate caution when interpreting how general the implications of this work are.

One final minor point -- can the results and experiments be clarified to explain whether any across-network comparisons are made in this work? Looking at Figure 1, it seems like all of the analyses compare hidden layer units within a single neural network -- i.e., hidden layer units are not compared across networks. Is this correct? If so, it may help to state this a bit more explicitly in section 4.

**Strengths And Weaknesses:**

The biggest strength of the paper is the systematic evaluation of different hidden unity similarity measures. This section was clearly motivated and well written.

I think the weakness of this work is that it only studies a very simple task (MNIST) and only a single architecture which is also very simple (relatively shallow, fully connected, feedforward network). Thus, it is not clear how general the results and conclusions of this work are. Do they still hold for convolutional networks? Do other architectural differences that are important in practice (e.g. batch norm, residual / skip connections, or choice of optimizer) affect the results? Similarly, do the results change with very large and complicated image datasets? This is particularly concerning because MNIST is a small dataset with exceedingly simple image features.

---

> ### Author Response · Authors · 2022-05-13
> **Response to reviewer 3**
>
> Thank you for your comments and for your careful reading of our work.
>
> > I would suggest that the authors add more experiments with (a) convolutional architectures, and (b) in more complex tasks (e.g. CIFAR-10 and CIFAR-100).
>
> This is a good suggestion, and the sentiment that our initial models were too simple was shared by the other reviewers.
>
> In the revision, we now include two additional experiments – “wider” and “deeper” networks. Our primary and most robust result is the misalignment between upstream and downstream measures of similarity. The effects of regularization on Q are somewhat secondary, and indeed we find some idiosyncracies across architectures (the wider (resp. deeper) models showed less (resp. more) extreme effects of dropout).
>
> We also began, but have not yet finished, investigating deep (6-block) residual and convolutional networks trained on CIFAR-10. Analyzing the similarity structure in these convolutional networks (in particular equations (2) and (4) in our paper) has turned out to be prohibitively slow using straightforward applications of autograd. We will continue to investigate methods to compute these more efficiently if deemed necessary above and beyond the two MNIST experiments already added, but unfortunately these results are out of scope for the present revision given the time constraints.
>
> > ​​Can the results and experiments be clarified to explain whether any across-network comparisons are made in this work?
>
> This depends somewhat on what you mean by “across-network.” In the original submission, we trained more than 250 separate models, and in the revision we now include additional models of the “wide” and “deep” architectures. But indeed, all clustering was done separately per layer, per model, and per similarity method. We therefore get a separate Q* value and P* cluster assignments per unique combination of {seed, layer, architecture, regularization, similarity type}.
>
> For the plots of Q* versus regularization, we show the mean ± standard error, averaging Q values across models, seeds, and layers. Supplemental Figures show finer-grained analyses, e.g. plotting separately for each layer. If each regularization value (each point on the x-axis)  is considered a separate network, then this is in a sense an across-network comparison. However, we suspect this is not what you meant?
>
> In the subsequent analyses of “alignment” of the clusters, alignment was also measured separately for each unique configuration as described above, then averaged to create the plot. To clarify this, we have now added the sentence, “Note that all comparisons between cluster assignments investigated in this section are performed separately per layer per model – cluster assignments are never compared for units across different models.”

---

> > ### Comment · Reviewer_g9XM · 2022-05-26
> > **Revision is satisfactory**
> >
> > I recommend publishing the paper in its current form.

---

### Review · Reviewer_fDiT · 2022-04-13

**Summary Of Contributions:**

This paper presents a study of “functional modularity” in neural networks. Here functional modules are obtained through clustering similar functioning units within a layer, and it is studied how:

1. the modules induced by 4 distinct similarity metrics (and their normalized versions) differ
2. how L1 regularization, L2 regularization, and dropout affect modularity

The first contribution essentially focuses on the differences between up-stream similarity metrics and down-stream similarity metrics. The main finding is that metrics belonging to these different classes induce a different modularization, suggesting that finding modular representations that reflect the structure of the inputs may be different from learning modular representations that reflect output structure.

The second contribution essentially measures the effect of these different regularizers. The main finding here is that dropout encourages a “false sense” of modularity due to encouraging different units to fulfill the same role (as to mitigate the effect of dropout). The validates a finding that was made previously regarding dropout benefiting clusterability of units (Filan et al., 2021). Regarding L1 and L2 regularization it is found that both only impact modularization in subtle ways.

The current study is limited to 3-layer MLPs with two hidden layers of 64 units that are trained on MNIST for handwritten digit classification.

**Broader Impact Concerns:**

No concerns.

**Requested Changes:**

### Changes critical to securing my recommendation for acceptance

Including evidence to support the claim that “[...] the main results we show here are likely robust”. This should focus on conducting similar experiments (but perhaps for a reduced set of metrics, regularizers):

* on different dataset (eg. CIFAR100 or ImageNet)
* using different network architectures (eg. CNN or at least varying width/depth for MNIST)

I strongly recommend exploring a more complex / different architecture also on a more complex dataset, since MNIST is so simple that adding more layers or making them wider (i.e. increasing capacity) probably won’t tell much about the effect of this in the general case (i.e. when having a network with the right amount of capacity for the task).

Some more investigation into the observation that up-stream and down-stream similarity measures yield very different modularizations. Perhaps, if a similar finding is obtained on other datasets (eg. as per the above) this won’t be needed, but else I would recommend conducting the following analysis. Basically, given a clustering obtained under one metric A, evaluate its modularity score under another metric B and see how far this is from the optimal value that can be achieved when clustering based on B. This should tell one whether two very different clusterings also yield modularity scores that are far apart. Currently it may well be (but please correct me in case my reasoning is wrong) that there are many possible different clusterings having a similar modularity score (close to optimal) and while different clusterings are found for different metrics, this is mainly a consequence of ending up of comparing two arbitrary candidates within these pools.


### Changes to strengthen the work

* I think it would be very interesting to consider similarity between units in different layers. This won’t require any new training, but just re-analysis of existing MLPs.

* To strengthen the results regarding L1 norm it may be useful to compare the sparsity of the layer directly with the achieved modularity score, i.e. verifying that L1 actually succeeds at making the layer more sparse and thus the hypothesis is correctly falsified.

### Minor changes

* In terms of presentation I may be better to remove table 1 to the appendix

* It would be good to comment on why L2 regularization was used for the L1 sweep and the L2 sweep? Seeing as how L2 regularization does influence modularity, this may have impacted results slightly.

* Regarding the conclusion about disentanglement not being useful for downstream tasks, please also see the opposite findings presented in Van Steenkiste, S., Locatello, F., Schmidhuber, J. and Bachem, O., 2019. Are disentangled representations helpful for abstract visual reasoning?. Advances in Neural Information Processing Systems, 32.

**Strengths And Weaknesses:**

The paper is well written, the methodology is clearly presented, and the authors do a good job at discussing various implications of design choices as well as connecting to different bodies of literature. The experimental design appears sound, and I don’t have doubts about the reproducibility of these results.

In my view the most significant finding is that up-stream and down-stream similarity measures yield very different modularizations. This is an interesting finding, but perhaps also somewhat strange given the simplicity of the task under consideration and the similarity of the metrics considered. I think it would be good to analyze this in more depth and I have some suggestions for this down below.

The findings regarding dropout, as well as L1 and L2 regularization are less significant in my view. I didn’t think that the hypothesis regarding L1 and L2 were very intuitive (since eg. it is unclear how much sparsity L1 actually induces), and even then, their effect on modularization is quite limited. Regarding dropout, perhaps one may conclude that the MLPs under consideration are biased towards having separate units fulfill only a single functionality, since the redundancy induced by dropouts does not smear out functionality, but rather creates duplicates. However, it seems premature to extend this claim beyond MNIST and the fixed MLP under consideration.

The limited setting is in general the main weakness of this work:

1. only a single toy task is considered (MNIST) and it is not clear whether this admits a modular solution in the first place
2. only a single network architecture is considered (3-layer MLP) and changes to the capacity of the model (depth or width) or other designs (eg. convolution based) are not considered
3. only similarity of units within the same layer is considered

Regarding (1) it is written that “While it is not obvious whether MNIST admits a meaningful “modular” solution, we expect that the main results we show here are likely robust”, however, no evidence is presented to support this claim.

Regarding (3) it is written that “[...] in principle modules could be scored in the same way after concatenating layers together.”, but no such effort is undertaken.

---

> ### Author Response · Authors · 2022-05-13
> **Response to reviewer 2**
>
> Thank you for your comments and your careful reading of our work.
>
> > Regarding (1) it is written that “While it is not obvious whether MNIST admits a meaningful “modular” solution, we expect that the main results we show here are likely robust”, however, no evidence is presented to support this claim.
>
> We have changed this particular line of text to read “While it is not obvious whether MNIST admits a meaningful ``modular'' solution, Experiments 2 and 3 suggest that our main result -- the misalignment between upstream and downstream definitions of neural similarity -- is fairly robust to width and depth. Still, it is plausible that upstream and downstream ways of thinking about neural representations will become better aligned in more structured tasks and larger models.”
>
> Our updated results are also in line with your reading of the significance of our various claims. Our primary and most robust result is the misalignment between upstream and downstream measures of similarity. The effects of regularization on Q are somewhat secondary, and do indeed show some idiosyncracies across architectures (the wider (resp. deeper) models showed less (resp. more) extreme effects of dropout).
>
> > Regarding (3) it is written that “[...] in principle modules could be scored in the same way after concatenating layers together.”, but no such effort is undertaken.
>
> We feel that this is out of scope for the present paper, since the current analyses already contain a large number of comparisons – 8 types of similarity, separately for each layer, and across 3 regularization types each with multiple strengths. In addition, we have now added two more architectures – one “wide” and one “deep.”
>
> That being said, we are open to running additional whole-network analyses and including them as supplemental results. Note also that analyzing one layer at a time has some precedence, in particular in the work of Watanabe and colleagues [1-3].
>
> > on different dataset (eg. CIFAR100 or ImageNet)
> > using different network architectures (eg. CNN or at least varying width/depth for MNIST)
>
> These are good suggestions, and in particular the sentiment to expand width and/or depth was shared by the other reviewers. Our revised manuscript now includes a wider (Experiment 2) and deeper (Experiment 3) version of the original network, trained on MNIST.
>
> We also began, but have not yet finished, investigating deep (6-block) residual and convolutional networks trained on CIFAR-10. Analyzing the similarity structure in these convolutional networks (in particular equations (2) and (4) in our paper) has turned out to be prohibitively slow using straightforward applications of autograd. We will continue to investigate methods to compute these more efficiently if deemed necessary above and beyond the two MNIST experiments already added, but unfortunately these results are out of scope for the present revision given the time constraints.
>
> > Basically, given a clustering obtained under one metric A, evaluate its modularity score under another metric B and see how far this is from the optimal value that can be achieved when clustering based on B. This should tell one whether two very different clusterings also yield modularity scores that are far apart.
>
> This is a great suggestion, and we have now included it under the name “transferability” (equation (9)) – results are shown in Supplemental Figure 12. Ultimately, we do see qualitative agreement with the original conclusions, namely that clusters found based on upstream methods do not “transfer” well to downstream similarity methods.
>
> > To strengthen the results regarding L1 norm it may be useful to compare the sparsity of the layer directly with the achieved modularity score, i.e. verifying that L1 actually succeeds at making the layer more sparse and thus the hypothesis is correctly falsified.
>
> The effect of L1 (and L2 and dropout) regularization on weight norms and sparsity is shown in Supplemental Figure S2 (formerly S1). To summarize, both L1 and L2 regularization have a large impact on sparsity, but neither has a large impact on Q (relative to dropout, that is, which has no effect on sparsity but a large effect on Q).
>
> > Minor changes…
>
> Thank you for these suggestions – all have been included in the revision.
>
> [1] Watanabe, C., Hiramatsu, K., & Kashino, K. (2019). Understanding community structure in layered neural networks. Neurocomputing, 367, 84–102. https://doi.org/10.1016/j.neucom.2019.08.020
> [2] Watanabe, C., Hiramatsu, K., & Kashino, K. (2018). Modular representation of layered neural networks. Neural Networks, 97, 62–73. https://doi.org/10.1016/j.neunet.2017.09.017
> [3] Watanabe, C. (2019). Interpreting Layered Neural Networks via Hierarchical Modular Representation. Communications in Computer and Information Science, 1143 CCIS, 376–388. https://doi.org/10.1007/978-3-030-36802-9_40

---

> > ### Comment · Reviewer_fDiT · 2022-05-20
> > **Author reply and current state of the paper is satisfactory**
> >
> > Thank you for the reply and revisions to the paper. The revised claim about robustness of these results, together with the additional insights gained from extending the MNIST study to wider and deeper networks (and validating transferability), address the majority of my concerns.
> >
> > Note that it was not my intention to suggest that an across-layers study was necessary to secure my recommendation, which is why I had listed it under "changes to strengthen the work". However, I do would have liked to see this study broadened to another dataset such as CIFAR100. While I strongly encourage to authors to continue pursuing this effort and incorporating these results in the paper (no matter the outcome and even if for a reduced setting), this won't be necessary to secure my recommendation at this time.
> >
> > To reiterate, I think this paper contributes and insightful study of modularity in feedforward neural networks. The main results regarding upstream vs downstream metrics, which appears robust, is highly surprising and to my knowledge a novel result. The broadened study to deeper and wider networks now allows for more nuanced statements about the contribution of various regularizers as is appropriate for an empirical study. I expect that this work will be of interest to many researchers interested in understanding modularity in neural networks.

---

### Review · Reviewer_TQWy · 2022-04-13

**Summary Of Contributions:**

The submission is an empirical study looking at functional similarity of hidden units. Specifically, The authors define define a series of (mostly linear) functional similarity measures and study how the structure of these functional similarities change with L1, L2 and dropout regularization. The authors motivate this study by appealing to the admittedly vague notion of "modularity". The empirical study is carried out via training fully connected networks on the MNIST dataset.

**Requested Changes:**

1. In order to make general claims, the authors need to show that their results are not sensitive to the width.

2. Given the comments regarding the vagueness of the definition of modularity above, I would suggest that the authors _either_ add significant evidence in support of their definitions of modularity (see below) _or_ remove all claims, explicit or implicit, regarding the effect of the hyper-parameters on modularity. This would include renaming the modularity score to something like structural similarity score.

If the authors would like to keep their claims regarding the change of modularity intact, I would recommend bolstering their definitions of modularity. Some suggestions would be:
* Comparison with some structural modularity measures and more robust arguments as to why it is the functional one that matters. (I am not sure if I agree with the statement that the structural modularity is useful only in promoting functional modularity. A structurally sparse network with a well tuned module system can act as a great inductive bias for learning and generalization.)
* Proof of principle examples where the modularity measures are applied to networks (hand-tuned or trained) which are generally agreed to be modular. The authors could even make a where/what model by concatenating networks pre-trained to be sensitive to where and what etc.

**Strengths And Weaknesses:**

### Strengths
* The paper is well written and the arguments are clear. The authors are generally upfront about the potential weaknesses of their approach. They also provide a good background on modularity literature and clarify that it is not a well-defined concept that everyone agrees on.
* The observations regarding the increased modularity score with increased dropout as well as the difference between the functional structure of the downstream vs upstream analysis are interesting.


### Weaknesses

* The effect of the width is not considered. It is possible that width 64 is too narrow to see the full effect of L1/L2 regularization. It is known that the behavior of training MLPs can be qualitatively different between narrow and wide limits.

Barring the above, the paper would be completely acceptable if the authors just talk about the functional structure of the latent space. However, it is the relationship between these concepts and modularity that is in my opinion the greatest weakness of the paper:
* The authors admit that this concept is not well-defined, however they continue to make claims about increased/decreased modularity. If the idea is not well-defined and the authors admit that they do are not making an attempt at constructing a well-defined definition, how can these statements be made, especially since the different measures of their modularity score do not always agree with each other.
* Given the vagueness of the notion of modularity, a comparison with at least one structural modularity measure would have been useful. Especially since it seems like the functional and structural modularity scores would behave completely opposite to each other in some cases. For example, in the submission, dropout increases the modularity score by increasing redundancy. However, I would suspect that increased dropout would decrease the structural modularity by decreasing the sparseness of the inter-layer weights.

---

> ### Author Response · Authors · 2022-05-13
> **Response to reviewer 1**
>
> Thank you again for your comments and suggestions, many of which we have taken into account in the revision.
>
> > The authors admit that this concept is not well-defined, however they continue to make claims about increased/decreased modularity.
>
> To clarify, we make claims about increased or decreased Q, where in the original papers by Newman and colleagues that introduced Q, it was referred to as a measure of the “modularity” of a network [1]. The difference between Q (modularity of a graph) and what we are really interested in (modularity as an system-design principle) is, as you note, an important one to keep in mind. Some past work has avoided this potential for confusion by referring to their quantitative proxy for modularity by another term such as “clusterability” [2], and we are open to making this kind of change – perhaps we could use “functional clusterability” (“structural similarity score” risks confusing ‘structural’ and ‘functional’ modules). To address this in the present version, we have added text in a number of places to emphasize that our use of “modularity” refers to its network-science definition, Q, and we further replaced the term “module” with “cluster” where there may have been some ambiguity.
>
> > A comparison with at least one structural modularity measure would have been useful. Especially since it seems like the functional and structural modularity scores would behave completely opposite to each other in some cases… I would suspect that increased dropout would decrease the structural modularity by decreasing the sparseness of the inter-layer weights.
>
> Comparing to a structural modularity method such as [2–4] is a nice idea for a control, but unfortunately one that we have not had time to implement for this revision. One existing similarity is worth highlighting – the method of Watanabe et al (2018) involves clustering units in a layer based on similarity of their incoming and outgoing weights. In our method, this is closely related to the Jacobian with respect to inputs or outputs – our “i-sens” and “o-sens” similarity measures. We do also report weight norms and sparseness as a function of dropout strength in Supplemental Figure S2 (formerly S1), and while we see a clear effect of higher dropout increasing the *norm* of the weights, we see little to no effect on their *sparsity* in any of our experiments.
>
> > In order to make general claims, the authors need to show that their results are not sensitive to the width.
>
> Thank you for this suggestion. We now include results for a “wider” model and a “deeper” model. Indeed, we found that the effect of dropout increasing redundancy was much less pronounced in our wide networks. However, what we feel our main result is – that “upstream” and “downstream” measures of function are different – remains in the wide networks (updated Figure 5).
>
> > I am not sure if I agree with the statement that the structural modularity is useful only in promoting functional modularity.
>
> We agree that structural approaches to modularity are both interesting and useful! In our original text, we wrote “In this work, we take the functional approach, based on the assumption that structural modularity is itself only useful insofar as it supports distinct functions…” We had intended for this to merely clarify the scope of our paper, with an emphasis on the word “assumption.”
>
> > Proof of principle examples where the modularity measures are applied to networks (hand-tuned or trained) which are generally agreed to be modular.
>
> This is a good suggestion, as it could better establish that our methods work on a “ground truth” kind of problem. However, designing such a ground-truth network with known modules is itself challenging. Designing and running this analysis was unfortunately out scope for the present revision, but we are open to including it if the reviewers agree that our methods need further verification. However, we emphasize that our end goal is not necessarily to produce a better module-detector, but to explore differences in the ways modularity – and more generally the ‘function’ of a unit in a network – have been approached in the past.
>
> [1] Newman, M. E. J. (2006). Modularity and community structure in networks. Proceedings of the National Academy of Sciences of the United States of America, 103(23), 8577–8582. https://doi.org/10.1073/pnas.0601602103
> [2] Filan, D., Casper, S., Hod, S., Wild, C., Critch, A., & Russell, S. (2021). Clusterability in Neural Networks. ArXiv. http://arxiv.org/abs/2103.03386
> [3] Csordás, R., van Steenkiste, S., & Schmidhuber, J. (2021). Are Neural Nets Modular? Inspecting Functional Modularity Through Differentiable Weight Masks. ICLR. http://arxiv.org/abs/2010.02066
> [4] Watanabe, C., Hiramatsu, K., & Kashino, K. (2018). Modular representation of layered neural networks. Neural Networks, 97, 62–73. https://doi.org/10.1016/j.neunet.2017.09.017

---

> > ### Comment · Reviewer_TQWy · 2022-05-23
> > **Response to rebuttal**
> >
> > I thank the authors for their response. The authors have addressed my concerns.

---

### Decision · Action_Editors · 2022-06-01

**Recommendation:** Accept as is

**Comment:**

With three reviewers recommending acceptance, I'm going to agree.

The primary criteria for including in TMLR is that the claims made in the paper are judged to be accurate and correct, for which I feel this review process helped with.  The reviewers had some concerns initially about the generality of the results and the authors responded with an extended set of experiments.  All three reviewers are now recommending acceptance after the changes.

In addition to the accuracy, I believe this paper is of interest to some in the TMLR community, in particular those who are interested in modularity.